# The Price of Sparsity: Sufficient Conditions for Sparse Recovery using Sparse and Sparsified Measurements

**Youssef Chaabouni**
Operations Research Center
Massachusetts Institute of Technology
Cambridge, MA 02139, USA
youss404@mit.edu

**David Gamarnik**
Operations Research Center
Massachusetts Institute of Technology
Cambridge, MA 02139, USA
gamarnik@mit.edu

## Abstract

We consider the problem of recovering the support of a sparse signal using noisy projections. While extensive work has been done on the dense measurement matrix setting, the sparse setting remains less explored. In this work, we establish sufficient conditions on the sample size for successful sparse recovery using sparse measurement matrices. Bringing together our result with previously known necessary conditions, we discover that, in the high-SNR regime where $ds/p \to +\infty$, sparse recovery using a sparse design exhibits a phase transition at an information-theoretic threshold of $n_{\text{INF}}^{\text{SP}} = \Theta\left(s \log\left(p/s\right) / \log\left(ds/p\right)\right)$ for the number of measurements, where $p$ denotes the signal dimension, $s$ the number of non-zero components of the signal, and $d$ the expected number of non-zero components per row of measurement. This expression makes the price of sparsity explicit: restricting each measurement to $d$ non-zeros inflates the required sample size by a factor of $\log s / \log\left(ds/p\right)$, revealing a precise trade-off between sampling complexity and measurement sparsity. Additionally, we examine the effect of sparsifying an originally dense measurement matrix on sparse signal recovery. We prove in the regime of $s = \alpha p$ and $d = \psi p$ with $\alpha, \psi \in (0, 1)$ and $\psi$ small that a sample of size $n_{\text{INF}}^{\text{Sp-ified}} = \Theta\left(p/\psi^2\right)$ is sufficient for recovery, subject to a certain uniform integrability conjecture, the proof of which is work in progress.

## 1 Introduction

In recent years, sparse signal recovery has gained significant attention, motivated by applications in compressive sensing [7, 2, 5]; signal denoising [3]; sparse regression [13]; data stream computing [4, 11, 14]; combinatorial group testing [6]; etc. Practical examples range from the single-pixel camera, MRI scanners and radar remote-sensing systems to error-correction schemes in digital communications and widely used image-compression formats [7, Chap. 1].

The problem can be formulated as follows. Consider a *signal* $\beta^\star \in \mathbb{R}^p$, unknown but a priori $s$-sparse for some given $s \le p$, a random *measurement matrix* $X \in \mathbb{R}^{n \times p}$ (also referred to as *design*, *features* or *data*) and a *noise* vector $Z \sim \mathcal{N}(0, \sigma^2 I_n)$, where $n \in \mathbb{N}$ denotes the *sample size* and $\sigma^2 > 0$ a fixed constant. An vector of *observations* (also known as *labels* or *annotations*) is given by:

$$Y := X\beta^\star + Z.$$

Sparse recovery refers to reconstructing $\beta^\star$ given $X$ and $Y$. Intuitively, this problem can be reduced to recovering the support $S^\star$ of $\beta^\star$, i.e. the set of indices of its non-zero components. In fact, once the support $S^\star$ is identified, the full signal can be estimated using the corresponding columns of $X$ via the closed-form maximum-likelihood estimator formula $\beta_{\text{MLE}} = X_{S^\star}^+ Y$, where $X_{S^\star}^+$ denotes the Moore-Penrose pseudoinverse of the submatrix formed by the columns of $X$ with indices in $S^\star$ [10].

39th Conference on Neural Information Processing Systems (NeurIPS 2025).

Traditionally, $X$ is considered to be a **dense** random matrix with sub-Gaussian entries. Previous works have shown that the complexity of the problem in terms of required sample size exhibits two phase transitions at two thresholds $n_{\text{INF}} < n_{\text{ALG}}$, yielding three regimes:

- $n < n_{\text{INF}}$: impossibility of recovery. Reeves et al. [16] show that if $n \leq (1 - \varepsilon) n_{\text{INF}}$ then the recovery of any fraction of the support of the signal is impossible.

- $n_{\text{INF}} < n < n_{\text{ALG}}$: super-polynomial complexity. Gamarnik and Zadik [8] show that if $n \geq (1 + \varepsilon) n_{\text{INF}}$ then the maximum-likelihood estimator (MLE) recovers the support of $\beta^\star$. Although solvable, the problem is widely believed to be algorithmically hard since the MLE exhibits an Overlap Gap Property (OGP) [8].

- $n > n_{\text{ALG}}$: polynomial-time recovery. Wainwright [18] shows that if $n \geq (1 + \varepsilon) n_{\text{ALG}}$ then the LASSO [17] succeeds in recovering the support of $\beta^\star$.

## 1.1 Sparse measurement setting

While dense matrices offer an optimal sample size, they are costly in terms of storage and computation. **Sparse** measurement matrices, where the number of non-zero entries per measurement vector scales significantly smaller than the signal dimension, mitigate these costs: they require significantly less storage and allow for more efficient computations, as matrix-vector multiplications and incremental updates can be performed faster. In addition, they enable efficient signal recovery algorithms by taking advantage of the structural properties of the problem [9]. However, this sparsity comes at the cost of increased sampling complexity [19]. This raises the following key question: How does measurement sparsity trade off with sampling complexity?

Some of the prior studies have explored this sparse measurement setting. Wang et al. [19] establish necessary conditions for sparse recovery for various measurement sparsity regimes. Let $d$ denote the expected number of non-zero components of a row of $X$. Their work reveals three regimes of behavior depending on $ds/p$, the expected number of non-zero components of $\beta^\star$ that align with non-zero components of a row of $X$. The three regimes are: $ds/p \to +\infty$, $ds/p = \tau$ for some constant $\tau > 0$, and $ds/p \to 0$. They show that in each regime, the number of samples $n$ must exceed a specific information-theoretic lower bound for any algorithm to reliably recover the signal's support. In particular, in the first regime, where $ds/p \to +\infty$, the necessary condition threshold of [19] is the same as the one of the dense case, while it increases dramatically in the third case, where $ds/p \to 0$. They work with entries rescaled so that $\text{Var}(X\beta^\star)$ matches the dense case, while we keep $\text{Var}(X_{ij}) = 1$. The settings are equivalent since any scaling of $X$ can be accounted for in $\beta^\star$.

In this work, we examine the opposite question: how many samples are enough to guarantee a reliable recovery? For simplicity, we assume the signal is binary, i.e. $\beta^\star \in \{0, 1\}^p$. Note that in this case, recovering the support is equivalent to recovering the signal. This assumption is very common in the literature [1, 16, 8]. Intuitively, detecting a component of size 1 is at least as hard as detecting a stronger component, so the resulting thresholds are representative of signals with non-zero entries bounded away from zero by 1, i.e. $\beta^\star \in \left\{ \beta \in \mathbb{R}^p : \|\beta\|_0 = s \text{ and } \min_{j \in [p] : \beta_j \neq 0} |\beta_j| \geq 1 \right\}$. Our first main result (Theorem 1) states that in the high signal-to-noise ratio (SNR) regime where $ds/p \to +\infty$, if the number of samples $n$ is larger than a threshold given by:

$$n_{\text{INF}}^{\text{SP}} = \Theta \left( \frac{s \log (p/s)}{\log (ds/p)} \right), \tag{1}$$

then the MLE asymptotically recovers the support of the signal. The proof uses large deviation techniques to bound the probability that a different support has a lower error than the true one, and a union bound over such supports. Bringing our result together with the necessary condition shown by Wang et al. [19], we reveal that the problem exhibits a phase transition – similar to the one known in the dense case – at the information-theoretic threshold $n_{\text{INF}}^{\text{SP}}$. In fact, if there exists a constant $\varepsilon > 0$ such that $n \leq (1 - \varepsilon) n_{\text{INF}}^{\text{SP}}$ then it is information-theoretically impossible to ensure a reliable recovery of the support of the signal, and if there exists a constant $\varepsilon > 0$ such that $n \geq (1 + \varepsilon) n_{\text{INF}}^{\text{SP}}$ then the MLE ensures a reliable recovery of the support. Our findings therefore answer the question of exactly how much data is needed for recovery. However, we note that the recovery that we show in the sufficiency statement holds in a weaker sense than the non-recovery in the necessity statement. We call the amount of additional observations in the sparsification setting compared to the dense one *price of sparsity*. Precisely, restricting each measurement to $d$ non-zeros inflates the required sample

size by a factor of $\Gamma = \Theta\left(\log s / \log\left(ds/p\right)\right)$, quantifying the sampling complexity vs. measurement sparsity trade-off.

Regarding the computational complexity, Omidiran and Wainwright [15] show that the LASSO performs as well in the sparse setting as in the dense setting, assuming a slow decay of sparsity. They show that, under some slow sparsity assumption, it is sufficient for the sample size $n$ to be larger than the algorithmic threshold of the dense setting discussed above, given by:

$$n_{\mathrm{ALG}} = 2s \log\left(p - s\right),$$

specifically for the LASSO to ensure a reliable polynomial time recovery of $\beta^\star$. Although the sparsity assumption under which this result holds allows for the density rate $d/p$ to go to 0 as $p \to +\infty$, it still doesn't allow the measurements to be very sparse. In fact, it requires that:

$$d/p = \omega\left(s^{-1/3}\right) \quad \text{and} \quad d/p = \omega\left(\left(\frac{\log\log\left(p - s\right)}{\log\left(p - s\right)}\right)^{1/3}\right). \tag{2}$$

This raises a question about what happens in a more sparse regime. Although we don't address the algorithmic threshold, i.e. the question of polynomial-time recovery, we discover herein a sufficient condition for recovery that allows for more sparsity, allowing for future investigation in this direction.

## 1.2 Sparsification

The applications of the signal recovery problem [7, Chapter 1] considered in this paper can be broadly categorized into two classes:

- Applications where $X$ is **designed**, e.g. involving signal compression and reconstruction.
- Applications where $X$ is **observed**, e.g. sparse regression, signal denoising, and error correction.

In light of this categorization, we note that measuring the trade-off between measurement sparsity and sampling complexity is particularly useful for the first class of problems. It provides practitioners with an exact description of how the measurement matrix should be designed, in terms of size and sparsity, to optimize the computational cost of signal recovery. However, this is rendered useless in the second class of problems when the measurement matrix is observed and dense. This motivates the second key question: given an initially dense measurement matrix, is there a way to make it sparse and still aim to recover the original signal?

This setting, in which the observations are generated with a dense measurement matrix, but the signal is recovered using a sparsified version of it, is closely related to the "missing covariates" or "missing-at-random" framework studied in high-dimensional statistics. Prior work by Loh and Wainwright [12], established algorithmic $\ell_2$-error bounds for regression under this model, assuming dense Gaussian designs and a constant missingness rate. Our analysis in Section 3 complements this line of research by focusing instead on information-theoretic support-recovery thresholds and deriving the precise sample complexity cost incurred by sparsification in this regime.

In our examination of the sparsification question, we focus on the linear sparsity and linear sparsification regime where $s = \alpha p$ and $d = \psi p$ for constant $\alpha, \psi \in (0, 1)$. Specifically, our second main result (Theorem 3) states that if the number of samples $n$ is larger than a threshold given by:

$$\frac{2h\left(\alpha\right)p}{\log\left(1 + \frac{\delta\psi^2}{(1-\psi)(2-\delta(1-\psi))}\right)}, \tag{3}$$

then the minimizer of the mean squared error (MSE) based on sparsified measurements and accordingly-rescaled observations asymptotically recovers the true support up to error fraction $\delta \in (0, 1)$. In particular, support recovery is possible for arbitrarily aggressive sparsification, i.e. arbitrarily small $\psi$. In the strong-sparsification regime where $\psi \to 0$, the sufficient threshold (3) effectively writes:

$$n_{\mathrm{INF}}^{\mathrm{Sp\text{-}ified}} = \Theta\left(\frac{p}{\psi^2}\right). \tag{4}$$

We call the amount of additional observations in the sparsification setting compared to the dense one *price of sparsification*. Unlike the price of sparsity, it is not due to the sparsity of the measurements but rather to a bias in the observations. We also interpret our result as providing an expression of the *sparsification budget*: the level up to which one could sparsify their data and still recover the true signal. Based on our result, this has order $\Theta\left(\sqrt{p/n}\right)$ when $n = \Omega\left(p\right)$.

### 1.3 Contributions

To the best of our knowledge, this work is the first to address the following:

1. Establish a necessary and sufficient condition for sparse recovery in the sparse setting.
2. Provide a sufficient condition for sparse recovery after sparsifying an originally dense design, conditional on a mild uniform-integrability conjecture.

### 1.4 Outline and Notations

We organize the rest of the paper as follows. Section 2 studies the sparse measurement setting. Section 3 examines recovery after sparsifying an originally dense measurement matrix. Section 4 concludes and sketches future work directions.

Throughout this document, we will use the following notations:

- We denote by $h(\cdot)$ the binary entropy: $h(x) = -x \log x - (1 - x) \log (1 - x)$, $x \in (0, 1)$.

- We call $\ell_0$-*norm* the number of non-zero coordinates of $x \in \mathbb{R}^d$, that is $\|x\|_0 := \sum_{i=1}^{d} \mathbb{1}(x_i \neq 0)$.

## 2 Sparse Recovery using Sparse Measurements

### 2.1 Setting

Let $n, p, s, d \in \mathbb{N}$ such that $d, s \leq p$. We define a sparse Gaussian matrix in $\mathbb{R}^{n \times p}$ as follows.

**Definition 2.1** (Sparse Gaussian matrix). We call $X = [X_{ij}]_{i \in [n], j \in [p]} \in \mathbb{R}^{n \times p}$ a sparse Gaussian matrix with parameter $d$ if for all $i \in [n], j \in [p]$ we have:

$$X_{ij} = B_{ij} N_{ij},$$

where $(B_{ij})_{i \in [n], j \in [p]} \overset{\text{i.i.d.}}{\sim} \text{Ber}(d/p)$ and $(N_{ij})_{i \in [n], j \in [p]} \overset{\text{i.i.d.}}{\sim} \mathcal{N}(0, 1)$ are mutually independent.

**Remark 2.1.** Note that $d$ is the expected number of non-zero components per row of $X$. In our setting, we think of $d$ as being of smaller order of magnitude than $p$. In particular, $d = o(p)$.

Let $X$ be a sparse Gaussian random matrix of parameter $d$, and $Z$ be a random vector in $\mathbb{R}^n$ such that $Z \sim \mathcal{N}(0, \sigma^2 I_n)$, with $\sigma > 0$ a fixed constant. Let $\beta^\star \in \{0, 1\}^p$ be a deterministic vector such that $\|\beta^\star\|_0 = s$. We define the random vector $Y$ as:

$$Y := X\beta^\star + Z. \tag{5}$$

Of particular interest is the *signal-to-noise ratio* (SNR), known to be an important quantity for characterizing the difficulty of sparse recovery problems [19, 16]. It's defined as follows:

$$\text{SNR} := \frac{\mathbb{E}\|X\beta^\star\|_2^2}{\mathbb{E}\|Z\|_2^2} = \frac{ds}{p\sigma^2}. \tag{6}$$

The maximum likelihood estimator (MLE) of $\beta^\star$ is defined by the random vector:

$$\hat{\beta} := \underset{\beta \in \{0,1\}^p, \|\beta\|_0 = s}{\arg\min} \|Y - X\beta\|_2^2. \tag{7}$$

We are interested in the minimum number of samples $n$ required so that the MLE (7) asymptotically recovers the true signal $\beta^\star$. We formalize the problem as follows.

### 2.2 Problem

We start by defining the support of a vector and the symmetric difference of supports.

**Definition 2.2** (Support). Let $u \in \mathbb{R}^p$. We call *support of* $u$ the set of indices of the non-zero components of $u$ and denote it $\text{Supp}(u) := \{i \in [p]: u_i \neq 0\}$. Note that $|\text{Supp}(u)| = \|u\|_0$.

**Definition 2.3** (Symmetric difference). We call *symmetric difference* between two sets $S_1$ and $S_2$ the set of elements in one but not the other and denote it $S_1 \triangle S_2 := (S_1 \cup S_2) \setminus (S_1 \cap S_2)$.

We formalize the problem we address below as follows: given an error tolerance $\delta \in (0, 1)$, we wish to determine the minimum number of samples $n$ as a function of $p$, $s$ and $d$ required so that:

$$\mathbb{P}_{X,Z}\left(\left|\text{Supp}\left(\beta^\star\right) \triangle \text{Supp}\left(\hat{\beta}\right)\right| < 2\delta s\right) \longrightarrow 1, \quad \text{as } n, p, s, d \rightarrow +\infty.$$

## 2.3 Results

Our first main result, Theorem 1, provides a sufficient condition on the sample size for reliable support recovery when using sparse measurements.

**Theorem 1** (Sufficient conditions for sparse recovery using sparse measurement matrices)**.** *Suppose $p, s, d \rightarrow +\infty$, $d = o(p)$ and $ds = \omega(p)$ (i.e. SNR $\rightarrow +\infty$). Let $\delta \in (0, 1)$. We consider two different regimes.*

1. *Assume $s = o(p)$. Let*

$$n^\star_{slin} := \frac{2s \log(p/s)}{\log(ds/p) + \log(\delta/(2\sigma^2))}.$$

   *If there exists $\varepsilon > 0$ such that $n \geq (1 + \varepsilon) n^\star_{slin}$, then the MLE $\hat{\beta}$ recovers $\beta^\star$ up to error $\delta$ w.h.p.:*

$$\mathbb{P}_{X,Z}\left(\left|\text{Supp}\left(\beta^\star\right) \triangle \text{Supp}\left(\hat{\beta}\right)\right| < 2\delta s\right) \geq 1 - \exp\left(-\varepsilon s \log(p/s) + o\left(s \log(p/s)\right)\right),$$

   *as $n, p, s, d \rightarrow +\infty$.*

2. *Assume there exists a constant $\alpha \in (0, 1)$ such that $s = \alpha p$. Let:*

$$n^\star_{lin} := \frac{2h(\alpha) p}{\log d + \log(\delta\alpha/(2\sigma^2))},$$

   *where $h(\cdot)$ denote the entropy function. If there exists $\varepsilon > 0$ such that $n \geq (1 + \varepsilon) n^\star_{lin}$, then the MLE $\hat{\beta}$ recovers $\beta^\star$ up to error $\delta$ w.h.p.:*

$$\mathbb{P}_{X,Z}\left(\left|\text{Supp}\left(\beta^\star\right) \triangle \text{Supp}\left(\hat{\beta}\right)\right| < 2\delta s\right) \geq 1 - \exp\left(-\varepsilon h(\alpha) p + o(p)\right),$$

   *as $n, p, s, d \rightarrow +\infty$.*

The proof of Theorem 1, given in appendix A.1, uses large deviation techniques to bound the probability of a high-error support to have a lower MSE than the true one, then a union bound over such supports. We give below a brief proof sketch of Theorem 1.

*Proof sketch.* Let $\mathcal{S}$ denote the set of supports of cardinality $s$ and $S^\star = \text{Supp}(\beta^\star)$. For any $S \in \mathcal{S}$, we denote by $\mathbb{1}_S$ the vector in $\{0, 1\}^p$ such that $[\mathbb{1}_S]_i = \mathbb{1}(i \in S)$ for all $i \in [p]$. We define the loss function $L$ over $\mathcal{S}$ such that $L(S) := \|Y - X\mathbb{1}_S\|_2^2$, so that $\text{Supp}\left(\hat{\beta}\right) = \arg\min_{S \in \mathcal{S}} L(S)$. As $p$ gets large, the event "$L(S) < L(S^\star)$" for any $S$ such that $|S \triangle S^\star| \geq 2\delta s$ is a *rare event*. The Chernoff bound yields:

$$\log \mathbb{P}\left(L(S) < L(S^\star)\right) \leq \frac{n}{2}\left(\log\left(\frac{2\sigma^2 p}{\delta ds}\right) + o(1)\right).$$

This step involves most of the technical work. Then, by union bound:

$$\mathbb{P}\left(\left|\text{Supp}\left(\hat{\beta}\right) \triangle S^\star\right| < 2\delta s\right) \geq 1 - \sum_{S\,:\,|S \triangle S^\star| \geq 2\delta s} \mathbb{P}\left(L(S) < L(S^\star)\right) \geq 1 - \binom{p}{s}\left(\frac{2\sigma^2 p}{\delta ds}\right)^{n/2}.$$

Solving for $n$, we obtain a critical threshold of $n^\star = \frac{\log\binom{p}{s}}{\log(ds/p) + \log(\delta/(2\sigma^2))}$. We conclude. $\quad\square$

Bringing together Theorem 1 with the necessary conditions shown by Wang et al. in [19], we obtain the following corollary.

**Corollary 2** (Information-theoretic phase transition)**.** *The sparse recovery in the sparse setting problem exhibits a phase transition at an information-theoretic threshold $n^{SP}_{INF}$.*

1. *In the first regime considered above, the expression of $n_{INF}^{SP}$ is given by:*

$$n_{INF}^{SP} := \frac{2s \log (p/s)}{\log (ds/p)}.$$

2. *In the second regime considered above, the expression of $n_{INF}^{SP}$ is given by:*

$$n_{INF}^{SP} := \frac{2h(\alpha) p}{\log d}.$$

*Specifically, in each of these regimes:*

(i) *If there exists $\varepsilon > 0$ such that $n \leq (1 - \varepsilon) n_{INF}^{SP}$ then, as $n, p, s, d \to +\infty$, there exists no decoder $g \colon \mathbb{R}^n \to \{\beta \in \{0, 1\}^p : \|\beta\|_0 = s\}$ such that:*

$$\max_{\beta^\star \in \{0,1\}^p, \|\beta^\star\|_0 = s} \mathbb{P}_{X,Z}\Big(g(Y) \neq Supp(\beta^\star)\Big) \to 0.$$

*In this sense, it is information-theoretically impossible to ensure an asymptotically reliable recovery.*

(ii) *If there exists $\varepsilon > 0$ such that $n \geq (1 + \varepsilon) n_{INF}^{SP}$, then as $n, p, s, d \to +\infty$:*

$$\frac{\left| Supp\left(\hat{\beta}\right) \triangle Supp(\beta^\star) \right|}{2s} \longrightarrow 0,$$

*in probability. In this sense, the MLE (7) ensures an asymptotically reliable recovery.*

The proof of Corollary 2 is given in appendix A.2. Statement (*i*) is due to Wang et al. [19], while statement (*ii*) follows from Theorem 1 and is the main contribution of this section.

**Remark 2.2** (Limitations). We note that the definition of "recovery" is not the same in statements (*i*) and (*ii*) of Corollary 2. The necessity statement (*i*) is about the impossibility a vanishing probability of exact equality of supports, which is stronger than the sufficiency statement of vanishing rescaled error in (*ii*). In particular, it is possible in theory that both statements hold simultaneously. To the best of our knowledge, the literature on the dense setting also suffers from this gap (see [8, 18]). While Reeves et al. [16] establish a clean information-theoretic phase transition in the dense setting (using the same definition of "recovery" for both bounds), their analysis considers different problem settings and convergence guarantees than ours.

We interpret Theorem 1 and Corollary 2 as follows.

- **Phase transition.** For simplicity, we only discuss the sublinear sparsity regime, defined by $s = o(p)$. Previous works on sparse recovery in the dense case ([16],[8]) have shown the existence of an information-theoretic threshold:

$$n_{INF} = \frac{2s \log (p/s)}{\log s}, \tag{8}$$

  at which the complexity of support recovery in terms of sample size exhibits a phase transition, where the recovery of any fraction of the support is impossible for $n \leq (1 - \varepsilon) n_{INF}$, and full recovery is guaranteed by the MLE for $n \geq (1 + \varepsilon) n_{INF}$. In light of this, we ask if the support recovery problem for the class of sparse measurement matrices described above exhibits a similar behavior. In Corollary 2, we show that indeed, it exhibits a similar phase transition at an information-theoretic threshold given by:

$$n_{INF}^{SP} = \frac{2s \log (p/s)}{\log (ds/p)}. \tag{9}$$

  In Table 1, we summarize these information-theoretic thresholds alongside known algorithmic thresholds for the sublinear sparsity regime $\big(s = o(p)\big)$, highlighting the comparison between dense and sparse measurements in the high-SNR setting.

- **Price of Sparsity.** In particular, we notice that $n_{INF}^{SP} \geq n_{INF}$. This confirms the intuition that sparse recovery requires more samples in the sparse measurement case: in fact, due to the sparsity of the measurement matrix, there is a low probability for the coefficient of a component of the signal at a given observation to be non-zero, and hence the need for a larger sample size for recovery. In light

Table 1: Comparison of Sample Complexity Thresholds for Sublinear Sparsity $\left(s = o\left(p\right)\right)$.

| Measurement | Info-Theoretic Necessary | Info-Theoretic Sufficient | Algorithmic Necessary | Algorithmic Sufficient |
|---|---|---|---|---|
| Dense | $\frac{2s\log(p/s)}{\log s}$ [16, 19] | $\frac{2s\log(p/s)}{\log s}$ [8, 16] | $2s\log(p-s)$ [8] | $2s\log(p-s)$ [18] |
| Sparse, high SNR | $\frac{2s\log(p/s)}{\log(ds/p)}$ [19] | $\frac{2s\log(p/s)}{\log(ds/p)}$ (Thm 1) | Unknown | $2s\log(p-s)$ [15]$^\dagger$ |

$^\dagger$Holds under the slow decay of sparsity assumption in [15], see (2).

of this, Corollary 2 is to be interpreted as providing an exact value for the *price of sparsity*, i.e. the extra amount of observations required in the sparse setting compared to the dense one, which is given by:

$$\Gamma := \frac{n_{\text{INF}}^{\text{SP}}}{n_{\text{INF}}} = \frac{\log s}{\log\left(ds/p\right)} > 1.$$

- **Dependence on $d$ and $s$.** Note that the expression of the price of sparsity heavily depends on the regimes of $d$ and $s$. The smaller the density rate $d/p$, the more "expensive" the desired sparsity of the measurements is, as suggested by the expression of $\Gamma$. In particular, $\Gamma$ could take any value in $(1, +\infty)$, depending on the regimes of $d$ and $s$ w.r.t. $p$.

  **Example 2.1.** Consider the setting where $s = p^\alpha$, $d = p^\beta$ with $\alpha, \beta \in (0, 1)$ such that $\alpha + \beta > 1$. Then $\Gamma = \alpha/\left(\alpha + \beta - 1\right) \in (1, +\infty)$, which goes to 1 when $\beta$ goes to 1 (low measurement sparsity), and goes to $+\infty$ when $\alpha$ is fixed and $\beta$ goes to $1 - \alpha$ (high measurement sparsity).

- **Measurement sparsity vs. Sampling complexity trade-off.** In this context, we conclude the existence of a measurement sparsity vs. sampling complexity trade-off, which can also be interpreted as a trade-off between sampling complexity and computational cost. We consider an example that highlights this trade-off.

  **Example 2.2.** Let $\psi\colon [e, +\infty) \longrightarrow [e, +\infty)$ such that $\psi\left(x\right) = x/\log x$. Consider two measurement matrices: $X_1$ a dense Gaussian in $\mathbb{R}^{n_1 \times p}$ and $X_2$ a sparse Gaussian $\mathbb{R}^{n_2 \times p}$ with only $d = \min\left(p^{o(1)}, \psi^{-1}\left(o\left(\psi\left(p\right)\right)\right)\right)$ expected non-zero entries per row; and an $s$-sparse signal $\beta^\star \in \mathbb{R}^p$, in the linear sparsity regime where $s = \alpha p$ for constant $\alpha \in (0, 1)$. On one hand, the number of samples required for reliable recovery raises from $n_1 = n_{\text{INF}} = \Theta\left(p/\log p\right)$ in the dense case to $n_2 = n_{\text{INF}}^{\text{SP}} = \Theta\left(p/\log d\right)$ in the sparse one. On the other hand the computational cost of recovery the support is better in the sparse case, as matrix-vector multiplication cost drops from $n_1 p = \Theta\left(p^2/\log p\right)$ to $n_2 d = \Theta\left(pd/\log d\right)$. This highlights a trade-off between sampling complexity and computational cost. Proofs of these statements are given in appendix A.3.

- **Allowing for more sparsity.** Note that the sparsity assumption under which Theorem 1 guarantees reliable recovery when there exists $\varepsilon > 0$ such that $n \geq \left(1 + \varepsilon\right) n_{\text{INF}}^{\text{SP}}$, which is:

$$ds/p \to +\infty, \tag{10}$$

is weaker than the sparsity assumption of the sufficient algorithmic threshold of Omidiran and Wainwright [15] which guarantees polynomial-time recovery if there exists $\varepsilon > 0$ such that $n \geq \left(1 + \varepsilon\right) n_{\text{ALG}}$. As they show, this holds under the assumption that:

$$\left(\frac{d}{p}\right)^3 \min\left\{s, \frac{\log\log\left(p - s\right)}{\log\left(p - s\right)}\right\} \to +\infty.$$

For example, when $s = \Theta\left(p\right)$, this requires that $d = \omega\left(p^{2/3}\right)$, while our result holds under the weaker assumption of $d = \omega\left(1\right)$. Our result allows for a significantly better sparsity, but this comes at the cost of potential super-polynomial computational complexity, since computing the MLE (7) is exponential-time in general.

## 3 Improving Sparse Recovery via Sparsification

### 3.1 Setting

Let $n, p, s, d \in \mathbb{N}$ and $\beta^\star \in \{0, 1\}^p$ $s$-sparse, defined as in Section 2.1. Let $X \in \mathbb{R}^{n \times p}$ such that $(X_{i,j})_{i \in [n], j \in [p]} \overset{\text{i.i.d.}}{\sim} \mathcal{N}\left(0, 1\right)$ and $Z \sim \mathcal{N}\left(0, \sigma^2 I_n\right)$, with $\sigma > 0$ constant. Let $Y := X\beta^\star + Z \in \mathbb{R}^n$.

Let $(B_{ij})_{i\in[n],j\in[p]} \overset{\text{i.i.d.}}{\sim} \text{Ber}(d/p)$. We define the following *sparsified* version of $X$:

$$\tilde{X} \in \mathbb{R}^{n\times p} \quad \text{such that} \quad \tilde{X}_{ij} := B_{ij}X_{ij}, \ \forall i \in [n], j \in [p]. \tag{11}$$

In addition, we define a rescaled version of $Y$ as follows:

$$\tilde{Y} := \frac{d}{p}Y \in \mathbb{R}^n. \tag{12}$$

An estimator of $\beta^\star$ is defined by the random vector:

$$\hat{\beta} := \underset{\beta\in\{0,1\}^p,\, \|\beta\|_0=s}{\arg\min} \left\| \tilde{Y} - \tilde{X}\beta \right\|_2^2. \tag{13}$$

That is, the observations are generated with a dense measurement matrix $(X)$, but the signal is recovered using a sparsification of that matrix $(\tilde{X})$. We formalize the problem we address below as follows: given an error tolerance $\delta \in (0,1)$, we wish to determine the minimum number of samples $n$ in terms of $p$, $s$ and $d$ required so that:

$$\mathbb{P}_{X,Z}\left(\left|\text{Supp}(\beta^\star)\triangle\text{Supp}(\hat{\beta})\right| < 2\delta s\right) \longrightarrow 1, \ \text{as } n,p,s,d \to +\infty.$$

## 3.2 Results

Our second main result, Theorem 3, provides a sufficient condition on the sample size for reliable support recovery after sparsifying an originally dense measurement matrix.

**Theorem 3** (Sufficient conditions for sparse recovery using sparsified measurements)**.** *Suppose that $p \to +\infty$ and there exist $\alpha, \psi \in (0,1)$ such that $s = \alpha p$ and $d = \psi p$. Let $\delta \in (0,1)$. Let:*

$$n^\star_{\text{Sp-ified}} := \frac{2h(\alpha)p}{\log\left(1 + \frac{\delta\psi^2}{(1-\psi)(2-\delta(1-\psi))}\right)}. \tag{14}$$

*If there exists $\varepsilon > 0$ such that $n \geq (1+\varepsilon)n^\star_{\text{Sp-ified}}$, then, under Conjecture B.1, $\hat{\beta}$ recovers $\beta^\star$ up to error $\delta$ w.h.p.:*

$$\mathbb{P}\left(\left|\text{Supp}(\beta^\star)\triangle\text{Supp}(\hat{\beta})\right| < 2\delta s\right) \geq 1 - \exp\left(-\varepsilon h(\alpha)p + o(p)\right),$$

*as $n,p,s,d \to +\infty$.*

**Remark 3.1** (Limitations)**.** We show that this result holds under a mild uniform-integrability conjecture. The statement of the conjecture is deferred to appendix B (Conjecture B.1) as it requires delving into technical details of the proof. We expect it to follow from standard concentration bounds as all relevant random terms are sub-Gaussian, and verification is work in progress.

The proof of Theorem 3 is given in appendix B. It follows the same general outline as the proof of Theorem 1, but is much more technically involved. In particular, deriving the large-deviation bound is harder because the MSE no longer decomposes neatly: $\tilde{Y}$ does not correspond to the true observations of the sparsified data but rather to a rescaling of the original observations (12). Instead of repeating a sketch here, we simply refer the reader back to the earlier proof sketch (see section 2.3). We interpret Theorem 3 as follows.

- **Arbitrary sparsification rate.** According to Theorem 3, support recovery is possible for arbitrarily aggressive sparsification, i.e. arbitrarily small $\psi$, provided a large enough sample size.

- **Strong-sparsification regime.** In the strong-sparsification regime where $\psi \to 0$, the denominator of $n^\star_{\text{Sp-ified}}$ in (14) is effectively $\delta\psi^2/(2-\delta)$, and hence the sufficient condition upper bound writes:

$$n^{\text{Sp-ified}}_{\text{INF}} = \frac{2(2-\delta)h(\alpha)p}{\delta\psi^2} = \Theta\left(\frac{p}{\psi^2}\right). \tag{15}$$

- **Price of Sparsification.** We interpret our result as providing a value for the *price of sparsification*, i.e. the extra amount of observations required due to the information loss resulting from sparsification. In the linear sparsity and strong-sparsification regime, it writes:

$$\Gamma_{\text{Sp-cation}} := \frac{n^{\text{Sp-ified}}_{\text{INF}}}{n_{\text{INF}}} = \frac{\Theta\left(p/\psi^2\right)}{\Theta\left(p/\log p\right)} = \Theta\left(\frac{\log p}{\psi^2}\right).$$

Unlike the intrinsically-sparse-observations setting studied in Section 2, this extra amount of required observations is *not* due to the sparsity of the measurements. In fact, one can check from (24) that in the linear sparsity regime where $d = \Theta(p)$, the price of sparsity is constant (i.e. $\Gamma = \Theta(1)$). Instead, the price of sparsification is due to a *bias in the observations* that we explain by the fact that the sparsified observations $\tilde{Y}$ were not obtained as noisy projections of the true signal as in the original model (5), but rather via a naïve rescaling of the original observations (12). By simply rescaling the observations we did not discard the information in $Y$ coming from the nullified components of $X$, hence introducing a bias.

- **Sparsification budget.** Given dense data and a fixed large enough sample size $n$, by up to how much could we sparsify the data and still get recovery? We call this the *sparsification budget*. According to our result (15), its expression in the strong-sparsification regime is given by:

$$\psi_{\text{budget}} = \Theta\left(\sqrt{p/n}\right).$$

In particular, the above expression only makes sense when $n = \Omega(p)$, below which Theorem 3 does not hold.

## 4 Conclusion and Future Work

In the first part of this paper, we have studied the problem of recovery of a binary signal $\beta^\star \in \{0, 1\}^p$ based on a sparse measurement matrix and noisy observations. Our main result is that, if the measurements have density rate $d/p$ then, assuming that the measurements and the signal are together not too sparse – in particular if $ds = \omega(p)$, i.e. high-SNR regime – it is possible to recover the true support asymptotically when the sample size is larger than the threshold given by Theorem 1. Combining our work with the necessary conditions of Wang el al. [19], we reveal an information-theoretic phase transition. The expression of the phase-transition threshold makes the *price of sparsity* explicit, revealing a precise trade-off between sampling complexity and measurement sparsity. In the following, we present a quick summary of all – to the best of our knowledge – results on sparse recovery in the sparse measurement setting, along with some future work directions.

- Information-theoretic threshold, sufficient conditions. In Theorem 1, we establish a sufficient condition for reliable recovery. However, this result is conditional on the high-SNR ($ds/p \to +\infty$) assumption. This raises the question of sufficient conditions when the measurements and signal are even more sparse.

- Informational-theoretic threshold, necessary conditions.
  - Wang et al. [19] have studied this problem. Their work reveals three regimes of behavior depending on the scaling of the expected number of non-zeros of $\beta^\star$ aligning with non-zeros of a row of $X$: $ds/p = \omega(1)$, $ds/p \to \tau$ for some $\tau > 0$, and $ds/p = o(1)$. For their model, where the variance of the non-zero components of $X$ scales in a way that makes the second moment of the projected signal $X\beta^\star$ remain the same as in the dense case: the necessary condition threshold is on the order of magnitude of the one in the dense case in the regime where $ds/p = \omega(1)$, while it increases dramatically in the regime where $ds/p = o(1)$.
  - In the dense setting, Reeves et al. [16] have shown that even the recovery of a fixed fraction of the support is information theoretically impossible below the phase transition threshold: this is what they call the *all-or-nothing* property. It would be interesting to extend this property to the sparse setting.

- Algorithmic threshold, sufficient conditions. Omidiran and Wainwright [15] have shown that under a low-sparsity assumption on the measurements, the sufficient condition of the dense setting, i.e. $n \geq (1 + \varepsilon) n_{\text{ALG}}$, is sufficient for the sparse setting as well. It would be interesting to explore the question of polynomial-time recovery in a stronger sparsity regime.

- Algorithmic threshold, necessary conditions. Although we cannot really hope to provide necessary conditions for polynomial-time recovery – unless conditionally on P $\neq$ NP – it would be interesting to provide a threshold under which the problem is believed to be algorithmically hard, as done by Gamarnik and Zadik [8] in the dense setting.

In the second part of this paper, we have studied the problem of recovering the signal based on *sparsified* – but originally dense – measurements and accordingly-rescaled observations. Our main

result is that, in the linear sparsity and linear sparsification regime where $s = \alpha p$ and $d = \psi p$ for constant $\alpha, \psi \in (0, 1)$, it is possible to recover the true support asymptotically when the sample size is larger than a threshold given by Theorem 3. This reveals that support recovery is possible for arbitrarily aggressive sparsification provided a large enough sample size, and provides an upper bound on the *price of sparsification*.

Nevertheless, we believe that the sparsification problem is infeasible for strong enough regimes of sparsification. In particular, we conjecture that the recovery is information-theoretically impossible no matter the sample size in the sub-linear sparsification regime where $d = o(p)$. We leave the exploration of this regime for future work.

## Acknowledgments and Disclosure of Funding

This work was supported by the National Science Foundation (NSF) under grant CISE-2233897. Youssef Chaabouni thanks Soulef Frikha for long-standing mathematical inspiration, and Mehdi Makni for many years of insightful discussions. He also thanks Marouane Nejjar, Panos Tsimpos, and Malo Lahogue for valuable feedback and exchanges.

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

# A  Sparse Recovery using Sparse Measurements: Proofs

## A.1  Proof of Theorem 1

*Proof of Theorem 1.* For any $i \in [n]$, we denote by $X_i := (X_{ij})_{j \in [p]}$, $B_i := (B_{ij})_{j \in [p]}$, $N_i := (N_{ij})_{j \in [p]}$. We denote by $S^\star := \mathrm{Supp}\,(\beta^\star)$ the support of $\beta^\star$. Let $\mathcal{S} := \{S \subset [p] : |S| = s\}$. We define the function:

$$L :\ \mathcal{S} \longrightarrow [0, +\infty)$$
$$S \longmapsto \|Y - X\mathbb{1}_S\|_2^2,$$

where $\mathbb{1}_S$ denotes the vector in $\{0,1\}^p$ such that $[\mathbb{1}_S]_j = \mathbb{1}\,(j \in S)$ for all $j \in [p]$. Note that, since $X$ and $Y$ are random, $L(S)$ is a random variable for every $S \in \mathcal{S}$. In addition, note that:

$$L(S) = \|Z\|_2^2 + \|X\,(\mathbb{1}_{S^\star} - \mathbb{1}_S)\|_2^2 + 2\langle Z, X\,(\mathbb{1}_{S^\star} - \mathbb{1}_S)\rangle \quad \forall\, S \in \mathcal{S},$$

and, in particular:

$$L(S^\star) = \|Z\|_2^2 = \sum_{i=1}^{n} Z_i^2.$$

Fix $S \in \mathcal{S}$ such that $M := |S \triangle S^\star|\,/2 \geq \delta s$, and let $U := S^\star \setminus S$, $V := S \setminus S^\star$. Note that $|U| = |V| = M$. We define:

$$\Delta := L(S) - L(S^\star).$$

**Proposition A.1.** *As $n, p, s, d \to +\infty$:*

$$\mathbb{P}\,(\Delta \leq 0) \leq \left(\frac{2\sigma^2 p}{\delta d s}\right)^{n/2} e^{o(n)}.$$

*Proof.* See appendix A.1.1.

Hence, we obtain:

$$\mathbb{P}\left(\|Y - X\mathbb{1}_S\|_2^2 \leq \|Y - X\mathbb{1}_{S^\star}\|_2^2\right) \leq \left(\frac{2\sigma^2 p}{\delta d s}\right)^{n/2} e^{o(n)}, \tag{16}$$

for any $S \in \{0,1\}^p$ such that $|S| = s$ and $|S \triangle S^\star| \geq 2\delta s$.

Using (16) and the union bound over the set of supports $S$ s.t. $|S \triangle S^\star| \geq 2\delta s$, we obtain:

$$\mathbb{P}_{X,Z}\left(\left|\mathrm{Supp}\,(\beta^\star) \triangle \mathrm{Supp}\,(\hat{\beta})\right| < 2\delta s\right)$$
$$\geq \mathbb{P}_{X,Z}\left(\|Y - X\mathbb{1}_S\|_2^2 > \|Y - X\mathbb{1}_{S^\star}\|_2^2,\, \forall\, S : |S \triangle S^\star| \geq 2\delta s\right)$$
$$= 1 - \mathbb{P}_{X,Z}\left(\exists\, S : |S \triangle S^\star| \geq 2\delta s,\, \|Y - X\mathbb{1}_S\|_2^2 \leq \|Y - X\mathbb{1}_{S^\star}\|_2^2\right)$$
$$\overset{\text{U.B.}}{\geq} 1 - \sum_{S\,:\,|S \triangle S^\star| \geq 2\delta s} \mathbb{P}_{X,Z}\left(\|Y - X\mathbb{1}_S\|_2^2 \leq \|Y - X\mathbb{1}_{S^\star}\|_2^2\right)$$
$$\geq 1 - \binom{p}{s}\left(\frac{2\sigma^2 p}{\delta d s}\right)^{n/2} e^{o(n)}.$$

**First regime:** $s = o\,(p)$**.**  Using the Corollary of Stirling:

$$\log \binom{p}{s} = s \log\,(p/s)\,(1 + o\,(1)),$$

in the RHS of the inequality above, we obtain:

$$\mathbb{P}_{X,Z}\left(\left|\mathrm{Supp}\,(\beta^\star) \triangle \mathrm{Supp}\,(\hat{\beta})\right| < 2\delta s\right)$$
$$\geq 1 - \exp\left[s \log\,(p/s)\,(1 + o\,(1)) - \frac{n}{2}\left(\log\left(\frac{\delta d s}{2\sigma^2 p}\right) + o\,(1)\right)\right].$$

Let $n^\star_{\text{slin}} := \frac{2s \log (p/s)}{\log (ds/p) + \log (\delta/(2\sigma^2))}$. Then if $n \geq (1 + \varepsilon) n^\star_{\text{slin}}$ for some constant $\varepsilon > 0$, we have:

$$\mathbb{P}_{X,Z} \left( \left| \text{Supp} (\beta^\star) \triangle \text{Supp} \left( \hat{\beta} \right) \right| < 2\delta s \right)$$

$$\geq 1 - \exp \left[ s \log (p/s) (1 + o(1)) - \frac{(1 + \varepsilon) n^\star_{\text{slin}}}{2} \left( \log \left( \frac{\delta ds}{2\sigma^2 p} \right) + o(1) \right) \right]$$

$$= 1 - \exp \left[ s \log (p/s) \left( -\varepsilon + o(1) - \frac{1 + \varepsilon}{\log (ds/p) + \log (\delta/(2\sigma^2))} o(1) \right) \right]$$

$$= 1 - \exp \left( s \log (p/s) \left( -\varepsilon + o(1) \right) \right).$$

Hence, as $n, p, s, d \to +\infty$:

$$\mathbb{P}_{X,Z} \left( \left| \text{Supp} (\beta^\star) \triangle \text{Supp} \left( \hat{\beta} \right) \right| < 2\delta s \right) \geq 1 - \exp \left( -\varepsilon s \log (p/s) + o\left( s \log (p/s) \right) \right) \longrightarrow 1.$$

**Second regime:** $s = \alpha p$, $\alpha \in (0, 1)$. Using the Corollary of Stirling:

$$\log \binom{p}{s} = ph(\alpha) (1 + o(1)),$$

we get:

$$\mathbb{P}_{X,Z} \left( \left| \text{Supp} (\beta^\star) \triangle \text{Supp} \left( \hat{\beta} \right) \right| < 2\delta s \right)$$

$$\geq 1 - \exp \left[ ph(\alpha) (1 + o(1)) - \frac{n}{2} \left( \log \left( \frac{\delta ds}{2\sigma^2 p} \right) + o(1) \right) \right].$$

Similarly to above, we take $n^\star_{\text{lin}} := \frac{2H(\alpha)p}{\log d + \log(\delta\alpha/(2\sigma^2))}$. If $n \geq (1 + \varepsilon) n^\star_{\text{lin}}$ for some constant $\varepsilon > 0$, then we obtain, as $p, s, d \to +\infty$:

$$\mathbb{P}_{X,Z} \left( \left| \text{Supp} (\beta^\star) \triangle \text{Supp} \left( \hat{\beta} \right) \right| < 2\delta s \right) \geq 1 - \exp \left( -\varepsilon h(\alpha) p + o(p) \right) \longrightarrow 1,$$

concluding the proof. $\qquad \square$

### A.1.1   Proof of Proposition A.1

*Proof of Proposition A.1.* We have:

$$\Delta := L(S) - L(S^\star)$$

$$= \| X (\mathbb{1}_{S^\star} - \mathbb{1}_S) \|_2^2 + 2 \langle Z, X (\mathbb{1}_{S^\star} - \mathbb{1}_S) \rangle$$

$$= \sum_{i=1}^n \langle X_i, \mathbb{1}_{S^\star} - \mathbb{1}_S \rangle^2 + 2 \sum_{i=1}^n Z_i \langle X_i, \mathbb{1}_{S^\star} - \mathbb{1}_S \rangle.$$

We denote by $(\Delta_i)_{i \in [n]}$ the terms of the sum in the above expression, that is:

$$\Delta_i := \langle X_i, \mathbb{1}_{S^\star} - \mathbb{1}_S \rangle^2 + 2 Z_i \langle X_i, \mathbb{1}_{S^\star} - \mathbb{1}_S \rangle.$$

Note that $(\Delta_i)_{i \in [n]}$ are i.i.d. and $\Delta = \sum_{i=1}^n \Delta_i$.

Now using the Chernoff bound:

$$\mathbb{P} (\Delta \leq 0) = \mathbb{P} (-\Delta \geq 0) = \inf_{\theta \geq 0} \mathbb{P} \left( e^{-\theta \Delta} \geq 1 \right) \leq \inf_{\theta \geq 0} M_{-\Delta_i} (\theta)^n. \qquad (17)$$

We now study the moment generating function of $-\Delta_i$, i.e. $M_{-\Delta_i} (\cdot)$. We have:

$$M_{-\Delta_i} (\theta) = \mathbb{E}_{X_i, Z_i} \left[ e^{-\theta \left[ \langle X_i, \mathbb{1}_{S^\star} - \mathbb{1}_S \rangle^2 + 2 Z_i \langle X_i, \mathbb{1}_{S^\star} - \mathbb{1}_S \rangle \right]} \right]$$

$$= \mathbb{E}_{X_i} \left[ e^{-\theta \langle X_i, \mathbb{1}_{S^\star} - \mathbb{1}_S \rangle^2} \mathbb{E}_{Z_i} \left[ e^{-2\theta Z_i \langle X_i, \mathbb{1}_{S^\star} - \mathbb{1}_S \rangle} \big| X_i \right] \right]$$

$$= \mathbb{E}_{X_i} \left[ e^{-\theta \langle X_i, \mathbb{1}_{S^\star} - \mathbb{1}_S \rangle^2} M_{Z_i | X_i} \left( -2\theta \langle X_i, \mathbb{1}_{S^\star} - \mathbb{1}_S \rangle \right) \right]$$

$$= \mathbb{E}_{X_i} \left[ e^{-\theta \langle X_i, \mathbb{1}_{S^\star} - \mathbb{1}_S \rangle^2} e^{\frac{1}{2} (-2\theta \langle X_i, \mathbb{1}_{S^\star} - \mathbb{1}_S \rangle)^2 \sigma^2} \right]$$

$$= \mathbb{E}_{X_i} \left[ e^{(-\theta + 2\theta^2 \sigma^2) \langle X_i, \mathbb{1}_{S^\star} - \mathbb{1}_S \rangle^2} \right]$$

$$= \mathbb{E}_{X_i} \left[ e^{(-\theta + 2\theta^2 \sigma^2) \left( \sum_{j \in U} X_{ij} - \sum_{j \in V} X_{ij} \right)^2} \right].$$

Plugging this expression into (17), we obtain:

$$\log \mathbb{P}\left(\Delta \leq 0\right) \leq n \inf_{\theta \geq 0} \log \mathbb{E}_{X_i}\left[e^{\left(-\theta + 2\theta^2\sigma^2\right)\left(\sum_{j \in U} X_{ij} - \sum_{j \in V} X_{ij}\right)^2}\right].$$

Studying the function $\theta \mapsto -\theta + 2\theta^2\sigma^2$ on $\mathbb{R}_{\geq 0}$ leads to the change of variable:

$$\inf_{\theta \geq 0} \log \mathbb{E}_{X_i}\left[e^{\left(-\theta + 2\theta^2\sigma^2\right)\left(\sum_{j \in U} X_{ij} - \sum_{j \in V} X_{ij}\right)^2}\right] \tag{18}$$

$$= \inf_{\theta \in (-\infty, 1/(8\sigma^2)]} \log \mathbb{E}_{X_i}\left[e^{-\theta\left(\sum_{j \in U} X_{ij} - \sum_{j \in V} X_{ij}\right)^2}\right], \tag{19}$$

One can check that the function $\theta \mapsto \log \mathbb{E}_{X_i}\left[e^{-\theta\left(\sum_{j \in U} X_{ij} - \sum_{j \in V} X_{ij}\right)^2}\right]$ is non-increasing over $(-\infty, 1/\left(8\sigma^2\right)]$. Hence (19) is equal to:

$$\log \mathbb{E}_{X_i}\left[e^{-\left(\sum_{j \in U} X_{ij} - \sum_{j \in V} X_{ij}\right)^2/\left(8\sigma^2\right)}\right].$$

Therefore, the Chernoff bound yields:

$$\log \mathbb{P}\left(\Delta \leq 0\right) \leq n \log \mathbb{E}_{X_i}\left[e^{-\left(\sum_{j \in U} X_{ij} - \sum_{j \in V} X_{ij}\right)^2/\left(8\sigma^2\right)}\right]. \tag{20}$$

Since $U \cap V = \varnothing$, we have:

$$\sum_{j \in U} X_{ij} - \sum_{j \in V} X_{ij} \stackrel{d}{=} \sum_{j \in U \cup V} X_{ij}.$$

Therefore:

$$\mathbb{E}\left[e^{-\left(\sum_{j \in U} X_{ij} - \sum_{j \in V} X_{ij}\right)^2/\left(8\sigma^2\right)}\right] = \mathbb{E}\left[e^{-\left(\sum_{j \in U \cup V} X_{ij}\right)^2/\left(8\sigma^2\right)}\right]$$

$$= \mathbb{E}\left[e^{-\left(\sum_{j \in U \cup V} B_{ij} N_{ij}\right)^2/\left(8\sigma^2\right)}\right]$$

$$= \mathbb{E}_{B_i}\left[\mathbb{E}_{N_i}\left[e^{-\left(\sum_{j \in U \cup V} B_{ij} N_{ij}\right)^2/\left(8\sigma^2\right)} \,\middle|\, B_i\right]\right]$$

$$= \mathbb{E}_{B_i}\left[\mathbb{E}_{N_i}\left[e^{-\left(\frac{\sum_{j \in U \cup V} B_{ij} N_{ij}}{\sqrt{\sum_{j \in U \cup V} B_{ij}}}\right)^2 \times \frac{\sum_{j \in U \cup V} B_{ij}}{8\sigma^2}} \,\middle|\, B_i\right]\right].$$

In addition, conditionally on $B_i$, we have:

$$\Gamma := \left(\frac{\sum_{j \in U \cup V} B_{ij} N_{ij}}{\sqrt{\sum_{j \in U \cup V} B_{ij}}}\right)^2 \stackrel{d}{=} \chi^2(1).$$

Its MGF is:

$$\mathbb{E}\left[e^{t\Gamma} \mid B_i\right] = M_{\Gamma|B_i}(t) = \frac{1}{\sqrt{1-2t}}, \text{ for } t < 1/2.$$

Hence:

$$\mathbb{E}_{B_i}\left[\mathbb{E}_{N_i}\left[e^{-\left(\frac{\sum_{j \in U \cup V} B_{ij} N_{ij}}{\sqrt{\sum_{j \in U \cup V} B_{ij}}}\right)^2 \times \frac{\sum_{j \in U \cup V} B_{ij}}{8\sigma^2}} \,\middle|\, B_i\right]\right] = \mathbb{E}_{B_i}\left[M_{\Gamma|B_i}\left(-\frac{\sum_{j \in U \cup V} B_{ij}}{8\sigma^2}\right)\right]$$

$$= \mathbb{E}_{B_i}\left[\frac{2\sigma}{\sqrt{4\sigma^2 + \sum_{j \in U \cup V} B_{ij}}}\right].$$

Let $U_{[\delta s]}$ and $V_{[\delta s]}$ respectively denote the sets of $\delta s$ smallest elements of $U$ and $V$. Note that this definition is legitimate since $|U| = |V| = M \geq \delta s$. Since $B_{ij} \geq 0$ for all $j \in U \cup V$, we have:

$$\sum_{j \in U \cup V} B_{ij} \geq \sum_{j \in U_{[\delta s]} \cup V_{[\delta s]}} B_{ij},$$

and hence:

$$\mathbb{E}_{B_i}\left[\frac{2\sigma}{\sqrt{4\sigma^2 + \sum_{j\in U\cup V} B_{ij}}}\right] \leq \mathbb{E}_{B_i}\left[\frac{2\sigma}{\sqrt{4\sigma^2 + \sum_{j\in U_{[\delta s]}\cup V_{[\delta s]}} B_{ij}}}\right].$$

Therefore, we get:

$$\mathbb{E}\left[e^{-\left(\sum_{j\in U} X_{ij} - \sum_{j\in V} X_{ij}\right)^2/(8\sigma^2)}\right] \leq \mathbb{E}_{B_i}\left[\frac{2\sigma}{\sqrt{4\sigma^2 + \sum_{j\in U_{[\delta s]}\cup V_{[\delta s]}} B_{ij}}}\right],$$

and plugging this into (20) yields:

$$\log\mathbb{P}\left(\Delta \leq 0\right) \leq n\log\left(\mathbb{E}_{B_i}\left[\frac{2\sigma}{\sqrt{4\sigma^2 + \sum_{j\in U_{[\delta s]}\cup V_{[\delta s]}} B_{ij}}}\right]\right). \tag{21}$$

Now note that, for any $i \in [n]$:

$$\sum_{j\in U_{[\delta s]}\cup V_{[\delta s]}} B_{ij} \stackrel{d}{=} \mathrm{Bin}\left(2\delta s, \frac{d}{p}\right),$$

In addition, since $d = o\left(p\right)$ and $ds/p \to +\infty$, we have:

**Lemma A.1.** *For any $i \in [n]$, the following holds: as $p \to +\infty$,*

$$\frac{1}{\sqrt{2\delta ds/p}}\left(\sum_{j\in U_{[\delta s]}\cup V_{[\delta s]}} B_{ij} - 2\delta ds/p\right) \xrightarrow{dist} \mathcal{N}\left(0,1\right).$$

*Proof.* The proof is a simple adaptation of the proof of the Central Limit Theorem. See appendix A.1.2.

Hence, there exists a choice of the underlying probability space and random variables $(B_{ij})_{i\in[n],j\in[p]}$ for which the random variable above converges almost surely and:

$$N := \lim_{p\to+\infty}\frac{1}{\sqrt{2\delta ds/p}}\left(\sum_{j\in U_{[\delta s]}\cup V_{[\delta s]}} B_{ij} - 2\delta ds/p\right) \sim \mathcal{N}\left(0,1\right).$$

The above yields:

$$\frac{2\sigma}{\sqrt{4\sigma^2 + \sum_{j\in U_{[\delta s]}\cup V_{[\delta s]}} B_{ij}}} = \frac{2\sigma}{\sqrt{4\sigma^2 + 2\delta ds/p + \sqrt{2\delta ds/p}\, N + o\left(\sqrt{ds/p}\right)}}.$$

Let:

$$V_p := \frac{2\sigma\sqrt{ds/p}}{\sqrt{4\sigma^2 + \sum_{j\in U_{[\delta s]}\cup V_{[\delta s]}} B_{ij}}}.$$

Then we have:

$$V_p = \frac{2\sigma\sqrt{ds/p}}{\sqrt{4\sigma^2 + 2\delta ds/p + \sqrt{2\delta ds/p}\, N + o\left(\sqrt{ds/p}\right)}}$$

$$= \frac{2\sigma}{\sqrt{2\delta + \frac{4\sigma^2}{ds/p} + \sqrt{\frac{2\delta}{ds/p}}\, N + o\left(1/\sqrt{ds/p}\right)}}.$$

Since $ds/p \to +\infty$ as $p \to +\infty$, the above yields:

$$V_p \longrightarrow \frac{2\sigma}{\sqrt{2\delta}} = \sqrt{\frac{2\sigma^2}{\delta}}.$$

In addition, we note the following:

**Lemma A.2.** $V_p$ *is uniformly integrable, that is: there exists $p' \in \mathbb{N}$ such that*

$$\lim_{T \to +\infty} \sup_{p \geq p'} \mathbb{E}\left[|V_p| \mathbb{1}_{\{|V_p|>T\}}\right] = 0.$$

*Proof.* See appendix A.1.3.

Therefore, we get:

$$\lim_{p \to +\infty} \mathbb{E}\left[V_p\right] = \sqrt{\frac{2\sigma^2}{\delta}}.$$

Hence, we write:

$$\mathbb{E}_{B_i}\left[\frac{2\sigma}{\sqrt{4\sigma^2 + \sum_{j \in U_{[\delta s]} \cup V_{[\delta s]}} B_{ij}}}\right] = \sqrt{\frac{2\sigma^2 p}{\delta ds}} + o\left(\left(\frac{ds}{p}\right)^{-1/2}\right).$$

We conclude:

$$
\begin{aligned}
\log \mathbb{P}\left(\Delta \leq 0\right) &\leq n \log \left(\mathbb{E}_{B_i}\left[\frac{2\sigma}{\sqrt{4\sigma^2 + \sum_{j \in U_{[\delta s]} \cup V_{[\delta s]}} B_{ij}}}\right]\right) \\
&= n \log \left(\sqrt{\frac{2\sigma^2 p}{\delta ds}} + o\left(\left(\frac{ds}{p}\right)^{-1/2}\right)\right) \\
&= n \left(\log \left(\sqrt{\frac{2\sigma^2 p}{\delta ds}}\right) + \log \left(1 + o\left(1\right)\right)\right) \\
&= n \left(\log \left(\sqrt{\frac{2\sigma^2 p}{\delta ds}}\right) + o\left(1\right)\right) \\
&= n \log \left(\sqrt{\frac{2\sigma^2 p}{\delta ds}}\right) + o(n),
\end{aligned}
$$

which yields the desired result:

$$\mathbb{P}\left(\Delta \leq 0\right) \leq \left(\frac{2\sigma^2 p}{\delta ds}\right)^{n/2} e^{o(n)}.$$

$\square$

## A.1.2   Proof of Lemma A.1

*Proof of Lemma A.1.* Let:

$$N_p := \frac{1}{\sqrt{2\delta ds/p}}\left(\sum_{j \in U_{[\delta s]} \cup V_{[\delta s]}} B_{ij} - 2\delta ds/p\right).$$

Its characteristic function writes:

$$
\begin{aligned}
&\Phi_{N_p}(t)\\
&= \mathbb{E}\left[e^{itN_p}\right]\\
&= e^{-it\sqrt{2\delta ds/p}}\left(\mathbb{E}\left[\exp\left(itB_{ij}/\sqrt{2\delta ds/p}\right)\right]\right)^{2\delta s}\\
&= e^{-it\sqrt{2\delta ds/p}}\left(1 - d/p + d/p\exp\left(it/\sqrt{2\delta ds/p}\right)\right)^{2\delta s}\\
&= e^{-it\sqrt{2\delta ds/p}}\left(1 + d/p\left(\exp\left(it/\sqrt{2\delta ds/p}\right) - 1\right)\right)^{2\delta s}\\
&= e^{-it\sqrt{2\delta ds/p}}\left(1 + d/p\left(\frac{it}{\sqrt{2\delta ds/p}} - \frac{t^2}{4\delta ds/p} + \mathcal{O}\left(\frac{-it^3}{(2\delta ds/p)^{3/2}}\right)\right)\right)^{2\delta s}\\
&= e^{-it\sqrt{2\delta ds/p}}\left(1 + d/p\left(\frac{it}{\sqrt{2\delta ds/p}} - \frac{t^2}{4\delta ds/p} + \mathcal{O}\left(\frac{1}{(2\delta ds/p)^{3/2}}\right)\right)\right)^{2\delta s}\\
&= \exp\left(-it\sqrt{2\delta ds/p} + 2\delta s\log\left(1 + d/p\left(\frac{it}{\sqrt{2\delta ds/p}} - \frac{t^2}{4\delta ds/p} + \mathcal{O}\left(\frac{1}{(2\delta ds/p)^{3/2}}\right)\right)\right)\right)\\
&= e^{-it\sqrt{2\delta ds/p}}\exp\left(2\delta s\left(d/p\left(\frac{it}{\sqrt{2\delta ds/p}} - \frac{t^2}{4\delta ds/p} + \mathcal{O}\left(\frac{1}{(2\delta ds/p)^{3/2}}\right)\right) + \mathcal{O}\left(\frac{d}{sp}\right)\right)\right)\\
&= \exp\left(-it\sqrt{2\delta ds/p} + it\sqrt{2\delta ds/p} - t^2/2 + \mathcal{O}\left(\frac{1}{\sqrt{2\delta ds/p}}\right) + \mathcal{O}\left(\frac{2\delta d}{p}\right)\right)\\
&= \exp\left(-t^2/2 + o\left(1\right)\right) \overset{p\to+\infty}{\longrightarrow} \exp\left(-t^2/2\right) = \Phi_{\mathcal{N}(0,1)}(t),
\end{aligned}
$$

for all $t \in \mathbb{R}$. The result follows. $\qquad\square$

### A.1.3  Proof of Lemma A.2

*Proof of Lemma A.2.* Fix $T > 2$. We have:

$$
V_p = \frac{2\sigma\sqrt{ds/p}}{\sqrt{4\sigma^2 + \sum_{i\in U_{[\delta s]}\cup V_{[\delta s]}} B_{ij}}} = \frac{2\sigma}{\sqrt{2\delta + \frac{4\sigma^2}{ds/p} + \sqrt{\frac{2\delta}{ds/p}}N + o\left(\frac{1}{\sqrt{ds/p}}\right)}}.
$$

By total probability, we have:

$$
\begin{aligned}
\mathbb{E}\left[|V_p|\,\mathbb{1}\left\{|V_p| > T\right\}\right] &= \mathbb{E}\left[|V_p|\,\mathbb{1}\left\{|V_p| > T\right\} \mid N \geq -(ds/p)^{1/4}\right]\mathbb{P}\left(N \geq -(ds/p)^{1/4}\right)\\
&\quad + \mathbb{E}\left[|V_p|\,\mathbb{1}\left\{|V_p| > T\right\} \mid N < -(ds/p)^{1/4}\right]\mathbb{P}\left(N < -(ds/p)^{1/4}\right).
\end{aligned}
$$

Let

$$
\Xi_1 := \mathbb{E}\left[|V_p|\,\mathbb{1}\left\{|V_p| > T\right\} \mid N \geq -(ds/p)^{1/4}\right]\mathbb{P}\left(N \geq -(ds/p)^{1/4}\right),
$$

and

$$
\Xi_2 := \mathbb{E}\left[|V_p|\,\mathbb{1}\left\{|V_p| > T\right\} \mid N < -(ds/p)^{1/4}\right]\mathbb{P}\left(N < -(ds/p)^{1/4}\right),
$$

so that $\mathbb{E}\left[|V_p|\,\mathbb{1}\left\{|V_p| > T\right\}\right] = \Xi_1 + \Xi_2$. We address the two terms separately.

- First case: $N \geq -(ds/p)^{1/4}$. We have:

$$
\frac{4\sigma^2}{ds/p} + \sqrt{\frac{2\delta}{ds/p}}N + o\left(\frac{1}{\sqrt{ds/p}}\right) \geq \frac{4\sigma^2}{ds/p} - \sqrt{2\delta}\left(\frac{ds}{p}\right)^{-1/4} + o\left(\frac{1}{\sqrt{ds/p}}\right).
$$

The RHS above goes to $0$ as $p \to +\infty$, and therefore it is $\geq -\delta$ for $p$ large enough, say for all $p \geq p_1$ for some $p_1 \in \mathbb{N}$. Hence we have:

$$|V_p| = V_p = \frac{2\sigma}{\sqrt{2\delta + \frac{4\sigma^2}{ds/p} + \sqrt{\frac{2\delta}{ds/p}}N + o\left(\frac{1}{\sqrt{ds/p}}\right)}} \leq \frac{2\sigma}{\sqrt{\delta}},$$

for all $p \geq p_1$. In addition, this implies:

$$\mathbb{1}\left\{|V_p| > T\right\} \leq \mathbb{1}\left\{\left(\frac{2\sigma}{\sqrt{\delta}}\right) > T\right\}.$$

In addition, we have:

$$\mathbb{P}\left(N \geq -(ds/p)^{1/4}\right) \leq 1,$$

which all together yield:

$$\Xi_1 \leq \frac{2\sigma}{\sqrt{\delta}} \mathbb{1}\left\{\left(\frac{2\sigma}{\sqrt{\delta}}\right) > T\right\}.$$

• Second case: $N < -(ds/p)^{1/4}$. We have:

$$|V_p| = V_p = \frac{2\sigma\sqrt{ds/p}}{\sqrt{4\sigma^2 + \sum_{i \in U_{[\delta s]} \cup V_{[\delta s]}} B_{ij}}} \leq \sqrt{ds/p},$$

which also implies:

$$\mathbb{1}\left\{|V_p| > T\right\} \leq \mathbb{1}\left\{\sqrt{ds/p} > T\right\} = \mathbb{1}\left\{T^2 < \frac{ds}{p}\right\}.$$

In addition, by Gaussian tail bounds we have:

$$\mathbb{P}\left(N < -(ds/p)^{1/4}\right) \leq e^{-\frac{\sqrt{ds/p}}{2}},$$

for $p$ large enough, say for all $p \geq p_2$ for some $p_2 \in \mathbb{N}$. Therefore:

$$\Xi_2 \leq \sqrt{ds/p}\, e^{-\frac{\sqrt{ds/p}}{2}} \mathbb{1}\left\{T^2 < \frac{ds}{p}\right\}.$$

Note that the term $\sqrt{ds/p}\, e^{-\frac{\sqrt{ds/p}}{2}}$ is decreasing as $ds/p$ increases (we have $ds/p > T^2 > 2$). Therefore:

$$\Xi_2 \leq T\, e^{-T/2} \mathbb{1}\left\{T^2 < \frac{ds}{p}\right\} \leq T\, e^{-T/2}.$$

We hence conclude that:

$$\mathbb{E}\left[|V_p| \mathbb{1}\left\{|V_p| > T\right\}\right] = \Xi_1 + \Xi_2 \leq \frac{2\sigma}{\sqrt{\delta}} \mathbb{1}\left\{\left(\frac{2\sigma}{\sqrt{\delta}}\right) > T\right\} + T\, e^{-T/2},$$

for all $p \geq p' := p_1 \vee p_2$. Therefore we have:

$$\sup_{p \geq p'} \mathbb{E}\left[|V_p| \frac{2\sigma}{\sqrt{\delta}} \mathbb{1}\left\{|V_p| > T\right\}\right] \leq \frac{2\sigma}{\sqrt{\delta}} \mathbb{1}\left\{\left(\frac{2\sigma}{\sqrt{\delta}}\right) > T\right\} + T\, e^{-T/2}.$$

This holds for any $T > 2$. Taking $T$ to $+\infty$, the result follows:

$$\lim_{T \to +\infty} \sup_{p \geq p'} \mathbb{E}\left[|V_p| \mathbb{1}\left\{|V_p| > T\right\}\right] = 0.$$

$\square$

## A.2 Proof of Corollary 2

*Proof of Corollary 2.* This proof relies on bringing together Theorem 1 with the following result from Wang et al. [19].

**Theorem 4** (Necessary condition for sparse ensembles, Corollary 2 of [19])**.** *Let the measurement matrix $X \in \mathbb{R}^{n \times p}$ be drawn with i.i.d. elements from the following distribution:*

$$X_{ij} = \begin{cases} \mathcal{N}\left(0, \frac{1}{\gamma}\right), & \text{w.p. } \gamma \\ 0, & \text{w.p. } 1 - \gamma \end{cases}, \quad \text{for all } i \in [n], \ j \in [p]; \tag{22}$$

*where $\gamma \in (0, 1]$. Let $\lambda > 0$ and*

$$\mathcal{C}_{p,s}(\lambda) := \left\{ \beta \in \mathbb{R}^p \ \middle| \ |Supp(\beta)| = s, \ \min_{i \in Supp(\beta)} |\beta_i| = \lambda \right\}.$$

*Assume that $\sigma^2 = 1$. Then, in the regime where $\gamma s \to +\infty$, a necessary condition for asymptotically reliable recovery over the signal class $\mathcal{C}_{p,s}(\lambda)$ is given by:*

$$n > \frac{\log \binom{p}{s} - 1}{\frac{1}{2} \log (1 + s\lambda^2)}.$$

Note that, while Theorem 4 is not stated on the exact same signal space $\mathcal{C}_{p,s}(\lambda)$ in [19] but rather on the larger:

$$\left\{ \beta \in \mathbb{R}^p \ \middle| \ |\text{Supp}(\beta)| = s, \ \min_{i \in \text{Supp}(\beta)} |\beta_i| \geq \lambda \right\},$$

it follows directly from their result on "restricted ensembles" where the signal components under consideration are set exactly to $\lambda$ (see section III.A. in [19]). We now proceed to prove Corollary 2.

(*i*) We show that (*i*) holds using Theorem 4, but this requires adapting our problem to the framework used by Wang et al. in [19]. In fact, note that the model used in Theorem 4 is different from the one we use in this paper, that we defined in (5). In their model, Wang et al. [19] rescale the non-zero components of $X$ by multiplying them by $1/\sqrt{\gamma}$, and require that the noise variance is $\sigma^2 = 1$. Therefore, we cannot directly use Theorem 4 in our setting. However, this difference can be fixed by a simple rescaling of our model. Note that our model defined by (5), where $X$ follows the sparse Gaussian distribution defined in Definition 2.1 and $Z \sim \mathcal{N}\left(0, \sigma^2 I_n\right)$, can be equivalently written as:

$$Y_0 = X_0 \beta_0^\star + Z_0,$$

where:

$$Y_0 := \frac{1}{\sigma} Y, \quad X_0 := \frac{1}{\sqrt{d/p}} X, \quad \beta_0^\star := \frac{\sqrt{d/p}}{\sigma} \beta^\star, \quad \text{and} \quad Z_0 := \frac{1}{\sigma} Z \sim \mathcal{N}\left(0, I_n\right).$$

Hence, it is a particular case of the model defined in (22), with:

$$\gamma := d/p \quad \text{and} \quad \lambda := \frac{\sqrt{\gamma}}{\sigma}.$$

In addition, the regime we consider of $d = \omega(p/s)$ corresponds exactly to the regime considered in Theorem 4 where $\gamma s \to +\infty$. Therefore, using Theorem 4, a necessary condition for an asymptotically reliable recovery of $\beta^\star$ in the considered regime is given by:

$$n > \frac{\log \binom{p}{s} - 1}{\frac{1}{2} \log (1 + s\lambda^2)} = \frac{2 \log \binom{p}{s} - 2}{\log (1 + ds/(p\sigma^2))} = \frac{2 \log \binom{p}{s}}{\log (ds/p)} (1 + o(1)). \tag{23}$$

**First regime:** $s = o(p)$.    Using the Corollary of Stirling:

$$\log \binom{p}{s} = s \log (p/s) (1 + o(1)),$$

the necessary condition (23) writes:

$$n > \frac{2s \log (p/s)}{\log (ds/p)} (1 + o(1)).$$

Let:

$$n_{\text{INF}}^{\text{SP}} := \frac{2s \log (p/s)}{\log (ds/p)}.$$

Assume there exists $\varepsilon > 0$ such that $n \leq (1 - \varepsilon) n_{\text{INF}}^{\text{SP}}$. We know that, for large enough $n, p, s, d$ we have:

$$n \leq (1 - \varepsilon) n_{\text{INF}}^{\text{SP}} < \frac{2s \log (p/s)}{\log (ds/p)} (1 + o(1)),$$

which contradicts the necessary condition. Therefore, it is information-theoretically impossible to ensure a reliable recovery of the support of $\beta^\star$.

**Second regime:** $s = \alpha p$.    Using the Corollary of Stirling:

$$\log \binom{p}{s} = p h(\alpha) (1 + o(1)),$$

the necessary condition (23) writes:

$$n > \frac{2h(\alpha) p}{\log d} (1 + o(1)).$$

Let:

$$n_{\text{INF}}^{\text{SP}} := \frac{2h(\alpha) p}{\log d}.$$

Similarly to above, we conclude that if there exists $\varepsilon > 0$ such that $n \leq (1 - \varepsilon) n_{\text{INF}}^{\text{SP}}$ then it is information-theoretically impossible to ensure a reliable recovery of the support of $\beta^\star$.

(*ii*) We show that (*ii*) holds using Theorem 1.

**First regime:** $s = o(p)$.    Let $\delta > 0$ and:

$$n_{\text{slin}}^\star := \frac{2s \log (p/s)}{\log (ds/p) + \log (\delta/(2\sigma^2))}.$$

Note that:

$$n_{\text{INF}}^{\text{SP}} = \frac{2s \log (p/s)}{\log (ds/p)} = n_{\text{slin}}^\star (1 + o(1)).$$

Assume there exists $\varepsilon$ such that $n \geq (1 + \varepsilon) n_{\text{INF}}^{\text{SP}}$. Then:

$$n \geq (1 + \varepsilon) (1 + o(1)) n_{\text{slin}}^\star = (1 + \varepsilon + o(1)) n_{\text{slin}}^\star \geq (1 + \varepsilon/2) n_{\text{slin}}^\star,$$

for $n, p, s, d$ large enough. Using Theorem 1, we have:

$$\mathbb{P}_{X,Z} \left( \frac{1}{2s} \left| \text{Supp} \left( \hat{\beta} \right) \triangle \text{Supp} (\beta^\star) \right| < \delta \right) \geq 1 - \exp \left( - \varepsilon s \log (p/s) + o(s \log (p/s)) \right).$$

Therefore, we obtain:

$$\mathbb{P}_{X,Z} \left( \frac{1}{2s} \left| \text{Supp} \left( \hat{\beta} \right) \triangle \text{Supp} (\beta^\star) \right| < \delta \right) \longrightarrow 1,$$

as $n, p, s, d \to +\infty$. Since this holds for all $\delta > 0$, we conclude:

$$\frac{\text{Supp} \left( \hat{\beta} \right) \triangle \text{Supp} (\beta^\star)}{2s} \longrightarrow 0,$$

in probability, as $n, p, s, d \to +\infty$.

**Second regime:** $s = \alpha p$. Let $\delta > 0$ and:

$$n_{\text{lin}}^{\star} := \frac{2h(\alpha)p}{\log d + \log(\delta\alpha/(2\sigma^2))}.$$

Note that:

$$n_{\text{INF}}^{\text{SP}} = \frac{2h(\alpha)p}{\log d} = n_{\text{lin}}^{\star}(1 + o(1)).$$

Assume there exists $\varepsilon$ such that $n \geq (1 + \varepsilon) n_{\text{INF}}^{\text{SP}}$. Then:

$$n \leq (1 + \varepsilon)(1 + o(1)) n_{\text{lin}}^{\star} = (1 + \varepsilon + o(1)) n_{\text{lin}}^{\star} \geq (1 + \varepsilon/2) n_{\text{lin}}^{\star},$$

for $n, p, s, d$ large enough. Using Theorem 1, we have:

$$\mathbb{P}_{X,Z}\left(\frac{1}{2s}\left|\text{Supp}\left(\hat{\beta}\right) \triangle \text{Supp}\left(\beta^{\star}\right)\right| < \delta\right) \geq 1 - \exp\left(-\varepsilon h(\alpha)p + o(p)\right).$$

Therefore, we obtain:

$$\mathbb{P}_{X,Z}\left(\frac{1}{2s}\left|\text{Supp}\left(\hat{\beta}\right) \triangle \text{Supp}\left(\beta^{\star}\right)\right| < \delta\right) \longrightarrow 1,$$

as $n, p, s, d \to +\infty$. Since this holds for all $\delta > 0$, we conclude:

$$\frac{\text{Supp}\left(\hat{\beta}\right) \triangle \text{Supp}\left(\beta^{\star}\right)}{2s} \longrightarrow 0,$$

in probability, as $n, p, s, d \to +\infty$.

$\square$

## A.3 Proof of Example 2.2

*Proof of Example 2.2.* We have $d = \min\left(p^{o(1)}, \psi^{-1}\left(o\left(\psi(p)\right)\right)\right)$, hence:

$$\begin{cases} \log d/\log p = o(1) \\ \psi(d)/\psi(p) = o(1) \end{cases}.$$

In addition:

$$n_1 = n_{\text{INF}} = \Theta(p/\log p), \quad n_2 = n_{\text{INF}}^{\text{SP}} = \Theta(p/\log d).$$

Therefore, we have:

$$\frac{n_2}{n_1} = \frac{\Theta(p/\log d)}{\Theta(p/\log p)} = \Theta\left(\frac{\log p}{\log d}\right) = \omega(1),$$

On one hand, the number of samples required for reliable recovery is better in the dense case:

$$n_1 = \Theta(p/\log p) = o(p/\log d) = o(n_2). \tag{24}$$

On the other hand, the computational cost of recovering the support is better in the sparse case. In fact, matrix-vector multiplications are made easier by sparsity: in the dense case, multiplying $X_1$ with a vector in $\mathbb{R}^p$ costs:

$$n_1 p = \Theta(p^2/\log p)$$

real number multiplications, while multiplying $X_2$ with a vector in $\mathbb{R}^p$ costs

$$n_2 d = \Theta(pd/\log d) = p\Theta(\psi(d)) = po(\psi(p)) = o(p^2/\log p) = o(n_1 p), \tag{25}$$

real number multiplications. This highlights the trade-off between sampling complexity and computational cost. $\square$

# B Improving Sparse Recovery via Sparsification: Proof of Theorem 3

*Proof of Theorem 3.* Let $\mathcal{S} := \{S \in [p]\colon |S| = s\}$. We define the error function:

$$L : \mathcal{S} \longrightarrow [0, +\infty)$$
$$S \longmapsto \left\| \tilde{Y} - \tilde{X}\mathbb{1}_S \right\|_2^2.$$

Note that:

$$
\begin{aligned}
L(S) &= \left\| \tilde{Y} - \tilde{X}\mathbb{1}_S \right\|_2^2 \\
&= \left\| \frac{d}{p}Y - \tilde{X}\mathbb{1}_S \right\|_2^2 \\
&= \left\| \frac{d}{p}(X\mathbb{1}_{S^\star} + Z) - \tilde{X}\mathbb{1}_S \right\|_2^2 \\
&= \left\| \frac{d}{p}X\mathbb{1}_{S^\star} - \tilde{X}\mathbb{1}_S + \frac{d}{p}Z \right\|_2^2.
\end{aligned}
$$

In particular, we have:

$$L(S^\star) = \left\| \left( \frac{d}{p}X - \tilde{X} \right)\mathbb{1}_{S^\star} + \frac{d}{p}Z \right\|_2^2.$$

Let $S \in \mathcal{S}$. We define $A(S) := S^\star \setminus S$, $B(S) := S \setminus S^\star$, $C(S) := S^\star \cap S$ and $M(S) := |A| = |B| = |S^\star \triangle S|/2$. Note that $|C| = s - M$. We are interested in bounding the following probability:

$$\mathbb{P}\left( L(S) - L(S^\star) \leq 0 \right).$$

Note that the term above only depends on $S$ through $M$. In fact, for any $S, S' \in \mathcal{S}$ such that $|S \triangle S^\star| = |S' \triangle S^\star|$ we have $L(S) \overset{d}{=} L(S')$. In light of this, we define a family of random variables $(\Delta(\eta))_{\eta s \in [s]}$ indexed by the rescaled symmetric difference $\eta = M/s$ as $L(S) - L(S^\star)$ for some $S$ such that $|S \triangle S^\star| = 2M$, that is:

For any $\eta \in \left\{ \dfrac{i}{s} : i \in [s] \right\}$, $\quad \Delta(\eta) := L(S) - L(S^\star)$, for some $S$ s.t. $|S \triangle S^\star| = 2\eta s$.

We extend the notation above to $\eta \in [0, 1]$ by defining $\Delta(\eta) := \Delta\left( \frac{\lceil \eta s \rceil}{s} \right)$.

**Proposition B.1.** *Let $\eta \in (0, 1]$ and $S \in \mathcal{S}$ such that $S \triangle S^\star = \lceil \eta s \rceil$. For any $\theta \in (0, +\infty)$, define:*

$$\gamma(\theta) := 2d^2\theta\sigma^2/p^2.$$

*Let $B \in \{0, 1\}^p$ denote a random vector such that $B_j \overset{i.i.d.}{\sim} Ber(d/p)$. For any $\theta \in (0, +\infty)$, we define the following random variables:*

$$\sigma_U^2(\theta, B) := \theta^2 \sum_{j \in A(S) \cup B(S)} B_j,$$

$$
\begin{aligned}
\sigma_V^2(\theta, B) := \frac{4d^2 s}{p^2} &+ \left( (1 + \gamma(\theta))^2 - \frac{4d(1 + \gamma(\theta))}{p} \right) \sum_{j \in A(S)} B_j \\
&+ (1 - \gamma(\theta))^2 \sum_{j \in B(S)} B_{ij} + 4\left( 1 - \frac{2d}{p} \right) \sum_{j \in C(S)} B_j,
\end{aligned}
$$

$$\mathrm{Cov}_{(U,V)}(\theta, B) := \theta \left( \left( 1 + \gamma(\theta) - \frac{2d}{p} \right) \sum_{j \in A(S)} B_j - (1 - \gamma(\theta)) \sum_{j \in B(S)} B_j \right),$$

$$\rho(\theta, B) := \frac{\mathrm{Cov}_{(U,V)}(\theta, B)}{\sigma_U(\theta, B)\sigma_V(\theta, B)}.$$

*In addition, for any $\theta \in (0, +\infty)$ we define the random variable:*

$$f(\theta, B) := \begin{cases} \dfrac{1}{\sigma_U(\theta,B)\sigma_V(\theta,B)\sqrt{\left(\frac{1}{\sigma_U(\theta,B)\sigma_V(\theta,B)} - \rho(\theta,B)\right)^2 - 1}} & \text{if } \frac{1}{\sigma_U(\theta,B)\sigma_V(\theta,B)} - \rho(\theta,B) > 1, \\ +\infty & \text{else.} \end{cases}$$

(26)

*Then we have:*

$$\mathbb{P}(\Delta(\eta) \leq 0) \leq \left(\inf_{\theta > 0} \mathbb{E}_B\big[f(\theta, B)\big]\right)^n$$

*Proof.* See section B.1. □

Now recall that $s$ and $d$ are both linear in $p$. In particular, there exist $\alpha, \psi \in (0, 1)$ such that:

$$\begin{cases} s = \alpha p \\ d = \psi p \end{cases}.$$

We have:

$$\gamma(\theta) = 2d^2\theta\sigma^2/p^2 = 2\sigma^2\psi^2\theta.$$

(27)

**Proposition B.2.** *We define:*

$$C^\star(\eta) := \frac{\psi}{(1-\psi)(2-\eta(1-\psi))} > 0.$$

*Define the function:*

$$\xi \colon \mathbb{N} \longrightarrow (0, +\infty)$$
$$p \longmapsto \frac{C^\star(\eta)}{2\alpha\psi p}.$$

*Consider the random vector $B \in \{0,1\}^p : B_j \overset{i.i.d.}{\sim} Ber(\psi)$. We define the random variable:*

$$H_p(B) = f(\xi(p), B).$$

*Then the following hold:*

    *(i) $H_p(B) \xrightarrow{a.s.} \sqrt{\frac{1}{1+\eta\psi C^\star(\eta)}}$ as $p \to +\infty$.*

    *(ii) $\mathbb{P}(\Delta(\eta) \leq 0) \leq \left(\mathbb{E}_B[H_p(B)]\right)^n$.*

*Proof.* See section B.2. □

**Conjecture B.1.** *The exists $p_0$ such that $(H_p(B))_{p \geq p_0}$ is uniformly integrable, that is:*

$$\lim_{T \to +\infty} \sup_{p \geq p_0} \mathbb{E}_B\Big[H_p(B)\mathbb{1}\big(H_p(B) > T\big)\Big] = 0.$$

Hence we have $H_p(B) \xrightarrow{a.s.} \sqrt{\frac{1}{1+\eta\psi C^\star(\eta)}}$, which by Conjecture B.1 yields:

$$\lim_{p \to +\infty} \mathbb{E}_B[H_p(B)] = \sqrt{\frac{1}{1 + \eta\psi C^\star(\eta)}}.$$

Therefore, we obtain:

$$\mathbb{E}_B[H_p(B)] = \sqrt{\frac{1}{1 + \eta\psi C^\star(\eta)}} + o(1).$$

Thus, by Proposition B.2 we have:

$$\frac{1}{n} \log \left( \mathbb{P} \left( \Delta \left( \eta \right) \leq 0 \right) \right) \leq \log \left( \mathbb{E}_B \left[ H_p \left( B \right) \right] \right)$$

$$= \log \left( \sqrt{\frac{1}{1 + \eta \psi C^\star \left( \eta \right)}} + o \left( 1 \right) \right)$$

$$= -\frac{1}{2} \log \left( 1 + \eta \psi C^\star \left( \eta \right) \right) + o \left( 1 \right).$$

Hence:

$$\mathbb{P} \left( \Delta \left( \eta \right) \leq 0 \right) \leq \left( 1 + \eta \psi C^\star \left( \eta \right) \right)^{-n/2} e^{o(n)}.$$

Taking the union bound:

$$\mathbb{P} \left( \left| \text{Supp} \left( \beta^\star \right) \triangle \text{Supp} \left( \hat{\beta} \right) \right| < 2 \delta s \right)$$

$$\geq \mathbb{P} \left( \left\| \tilde{Y} - \tilde{X} \mathbb{1}_S \right\|_2^2 > \left\| \tilde{Y} - \tilde{X} \mathbb{1}_{S^\star} \right\|_2^2, \ \forall S \colon \left| S \triangle S^\star \right| \geq 2 \delta s \right)$$

$$= 1 - \mathbb{P} \left( \exists S \colon \left| S \triangle S^\star \right| \geq 2 \delta s, \left\| \tilde{Y} - \tilde{X} \mathbb{1}_S \right\|_2^2 \leq \left\| \tilde{Y} - \tilde{X} \mathbb{1}_{S^\star} \right\|_2^2 \right)$$

$$= 1 - \sum_{S \colon |S \triangle S^\star| \geq 2 \delta s} \mathbb{P} \left( \left\| \tilde{Y} - \tilde{X} \mathbb{1}_S \right\|_2^2 \leq \left\| \tilde{Y} - \tilde{X} \mathbb{1}_{S^\star} \right\|_2^2 \right)$$

$$= 1 - \sum_{\eta \in \{i/s \colon i \in [s]\} \colon \eta \geq \delta} \ \sum_{S \colon |S \triangle S^\star| = 2 \eta s} \mathbb{P} \left( L \left( S \right) \leq L \left( S^\star \right) \right)$$

$$= 1 - \sum_{\eta \in \{i/s \colon i \in [s]\} \colon \eta \geq \delta} \ \sum_{S \colon |S \triangle S^\star| = 2 \eta s} \mathbb{P} \left( \Delta \left( \eta \right) \leq 0 \right)$$

$$\geq 1 - \binom{p}{s} \sum_{\eta \in \{i/s \colon i \in [s]\} \colon \eta \geq \delta} \left( 1 + \eta \psi C^\star \left( \eta \right) \right)^{-n/2} e^{o(n)}.$$

In addition, we have:

$$\sum_{\eta \in \{i/s \colon i \in [s]\} \colon \eta \geq \delta} \left( 1 + \eta \psi C^\star \left( \eta \right) \right)^{-n/2} \leq \sum_{\eta \in \{i/s \colon i \in [s]\} \colon \eta \geq \delta} \left( 1 + \delta \psi C^\star \left( \delta \right) \right)^{-n/2} \qquad (28)$$

$$\leq s \left( 1 + \delta \psi C^\star \left( \delta \right) \right)^{-n/2}. \qquad (29)$$

where (28) holds because the term inside the sum is non-increasing in $\eta$, and (29) holds because the cardinality of the index set of the sum is upper bounded by $s$. Therefore:

$$\mathbb{P} \left( \left| \text{Supp} \left( \beta^\star \right) \triangle \text{Supp} \left( \hat{\beta} \right) \right| < 2 \delta s \right)$$

$$\geq 1 - \binom{p}{s} s \left( 1 + \delta \psi C^\star \left( \delta \right) \right)^{-n/2} e^{o(n)}$$

$$= 1 - \exp \left[ p h \left( \alpha \right) \left( 1 + o \left( 1 \right) \right) + \log s - n \left( \frac{1}{2} \log \left( 1 + \delta \psi C^\star \left( \delta \right) \right) + o \left( 1 \right) \right) \right]$$

$$= 1 - \exp \left[ p h \left( \alpha \right) \left( 1 + o \left( 1 \right) \right) - n \left( \frac{1}{2} \log \left( 1 + \delta \psi C^\star \left( \delta \right) \right) + o \left( 1 \right) \right) \right].$$

We define:

$$n^\star := \frac{2 h \left( \alpha \right) p}{\log \left( 1 + \delta \psi C^\star \left( \delta \right) \right)}.$$

Let $\varepsilon > 0$. Then if $n \geq (1 + \varepsilon) n^\star$, we have:

$$\mathbb{P}\left(\left|\operatorname{Supp}\left(\beta^\star\right) \triangle \operatorname{Supp}\left(\hat{\beta}\right)\right| < 2\delta s\right)$$

$$\geq 1 - \exp\left[ph\left(\alpha\right)\left(1 + o\left(1\right)\right) - n\left(\frac{1}{2}\log\left(1 + \delta\psi C^\star\left(\delta\right)\right) + o\left(1\right)\right)\right]$$

$$= 1 - \exp\left[ph\left(\alpha\right)\left(-\varepsilon + o\left(1\right)\right)\right]$$

$$\xrightarrow{p \to +\infty} 1.$$

$$\square$$

## B.1 Proof of Proposition B.1

*Proof of Proposition B.1.* Since $\eta$ and $S$ are fixed, we will simplify notations in this proof as follows: we will write $\Delta$ for $\Delta\left(\eta\right)$, and $A, B, C, M$ respectively for $A\left(S\right), B\left(S\right), C\left(S\right), M\left(S\right)$. We have:

$$\Delta = L(S) - L(S^\star)$$

$$= \left\|\frac{d}{p}X\mathbb{1}_{S^\star} - \tilde{X}\mathbb{1}_S + \frac{d}{p}Z\right\|_2^2 - \left\|\left(\frac{d}{p}X - \tilde{X}\right)\mathbb{1}_{S^\star} + \frac{d}{p}Z\right\|_2^2$$

$$= \left\|\tilde{X}\mathbb{1}_S\right\|_2^2 - \left\|\tilde{X}\mathbb{1}_{S^\star}\right\|_2^2 + 2\frac{d}{p}\left\langle X\mathbb{1}_{S^\star} + Z, \tilde{X}\left(\mathbb{1}_{S^\star} - \mathbb{1}_S\right)\right\rangle$$

$$= \sum_{i=1}^n \langle \tilde{X}_i, \mathbb{1}_S\rangle^2 - \sum_{i=1}^n \langle \tilde{X}_i, \mathbb{1}_{S^\star}\rangle^2 + 2\frac{d}{p}\sum_{i=1}^n \left(\langle X_i, \mathbb{1}_{S^\star}\rangle + Z_i\right)\left(\langle \tilde{X}_i, \mathbb{1}_{S^\star}\rangle - \langle \tilde{X}_i, \mathbb{1}_S\rangle\right)$$

$$= \sum_{i=1}^n \left(\langle \tilde{X}_i, \mathbb{1}_S\rangle^2 - \langle \tilde{X}_i, \mathbb{1}_{S^\star}\rangle^2 + 2\frac{d}{p}\left(\langle X_i, \mathbb{1}_{S^\star}\rangle + Z_i\right)\left(\langle \tilde{X}_i, \mathbb{1}_{S^\star}\rangle - \langle \tilde{X}_i, \mathbb{1}_S\rangle\right)\right)$$

Let:

$$\Delta_i := \langle \tilde{X}_i, \mathbb{1}_S\rangle^2 - \langle \tilde{X}_i, \mathbb{1}_{S^\star}\rangle^2 + 2\frac{d}{p}\left(\langle X_i, \mathbb{1}_{S^\star}\rangle + Z_i\right)\left(\langle \tilde{X}_i, \mathbb{1}_{S^\star}\rangle - \langle \tilde{X}_i, \mathbb{1}_S\rangle\right).$$

So that $\Delta = \sum_{i=1}^n \Delta_i$. Now using the Chernoff bound:

$$\mathbb{P}\left(\Delta \leq 0\right) = \mathbb{P}\left(-\Delta \geq 0\right) = \inf_{\theta \geq 0} \mathbb{P}\left(e^{-\theta\Delta} \geq 1\right) \leq \inf_{\theta \geq 0} M_{-\Delta_i}(\theta)^n. \tag{30}$$

We have:

$$\Delta_i = \langle \tilde{X}_i, \mathbb{1}_S\rangle^2 - \langle \tilde{X}_i, \mathbb{1}_{S^\star}\rangle^2 + 2\frac{d}{p}\left(\langle X_i, \mathbb{1}_{S^\star}\rangle + Z_i\right)\left(\langle \tilde{X}_i, \mathbb{1}_{S^\star}\rangle - \langle \tilde{X}_i, \mathbb{1}_S\rangle\right)$$

$$= \left(\sum_{j \in S} \tilde{X}_{ij}\right)^2 - \left(\sum_{j \in S^\star} \tilde{X}_{ij}\right)^2 + 2\frac{d}{p}\left(\sum_{j \in S^\star} X_{ij} + Z_i\right)\left(\sum_{j \in S^\star} \tilde{X}_{ij} - \sum_{j \in S} \tilde{X}_{ij}\right)$$

$$= -\left(\sum_{j \in S^\star} \tilde{X}_{ij} - \sum_{j \in S} \tilde{X}_{ij}\right)\left(\sum_{j \in S^\star} \tilde{X}_{ij} + \sum_{j \in S} \tilde{X}_{ij} - 2\frac{d}{p}\sum_{j \in S^\star} X_{ij}\right)$$

$$+ 2\frac{d}{p}Z_i\left(\sum_{j \in S^\star} \tilde{X}_{ij} - \sum_{j \in S} \tilde{X}_{ij}\right)$$

Then:

$$\Delta_i = -\left(\sum_{j \in A} \tilde{X}_{ij} - \sum_{j \in B} \tilde{X}_{ij}\right)\left(\sum_{j \in A} \tilde{X}_{ij} + \sum_{j \in B} \tilde{X}_{ij} + 2\sum_{j \in C} \tilde{X}_{ij} - 2\frac{d}{p}\sum_{j \in S^\star} X_{ij}\right)$$

$$+ 2\frac{d}{p}Z_i\left(\sum_{j \in A} \tilde{X}_{ij} - \sum_{j \in B} \tilde{X}_{ij}\right).$$

Let:

$$K_i := -\left(\sum_{j\in A}\tilde{X}_{ij} - \sum_{j\in B}\tilde{X}_{ij}\right)\left(\sum_{j\in A(S)}\tilde{X}_{ij} + \sum_{j\in B(S)}\tilde{X}_{ij} + 2\sum_{j\in C(S)}\tilde{X}_{ij} - 2\frac{d}{p}\sum_{j\in S^\star}X_{ij},\right)$$

so that:

$$\Delta_i = K_i + 2\frac{d}{p}\,Z_i\left(\sum_{j\in A}\tilde{X}_{ij} - \sum_{j\in B}\tilde{X}_{ij}\right).$$

The MGF of $-\Delta_i$ can be expressed as:

$$M_{-\Delta_i}(\theta) = \mathbb{E}\left[\exp\left(-\Delta_i\theta\right)\right]$$

$$= \mathbb{E}_{X_i,B_i}\left[\mathbb{E}_{Z_i}\left[\exp\left(-\theta\left(K_i + 2\frac{d}{p}\,Z_i\left(\sum_{j\in A}\tilde{X}_{ij} - \sum_{j\in B}\tilde{X}_{ij}\right)\right)\right)\middle|X_i,B_i\right]\right]$$

$$= \mathbb{E}_{X_i,B_i}\left[e^{-\theta K_i}\mathbb{E}_{Z_i}\left[\exp\left(-\frac{2d\theta}{p}\,Z_i\left(\sum_{j\in A}\tilde{X}_{ij} - \sum_{j\in B}\tilde{X}_{ij}\right)\right)\middle|X_i,B_i\right]\right]$$

$$= \mathbb{E}_{X_i,B_i}\left[e^{-\theta K_i}M_{Z_i\,|\,X_i,B_i}\left(-\frac{2d\theta}{p}\left(\sum_{j\in A}\tilde{X}_{ij} - \sum_{j\in B}\tilde{X}_{ij}\right)\right)\right]$$

$$= \mathbb{E}_{X_i,B_i}\left[e^{-\theta K_i}\exp\left(\frac{1}{2}\left(-\frac{2d\theta}{p}\left(\sum_{j\in A}\tilde{X}_{ij} - \sum_{j\in B}\tilde{X}_{ij}\right)\right)^2\sigma^2\right)\right]$$

$$= \mathbb{E}_{X_i,B_i}\left[\exp\left(-\theta K_i + \frac{2d^2\theta^2\sigma^2}{p^2}\left(\sum_{j\in A}\tilde{X}_{ij} - \sum_{j\in B}\tilde{X}_{ij}\right)^2\right)\right]$$

Thus, by the tower rule:

$$M_{-\Delta_i}(\theta) = \mathbb{E}_{B_i}\left[\mathbb{E}_{X_i}\left[\exp\left(-\theta K_i + \frac{2d^2\theta^2\sigma^2}{p^2}\left(\sum_{j\in A}\tilde{X}_{ij} - \sum_{j\in B}\tilde{X}_{ij}\right)^2\right)\middle|B_i\right]\right]. \qquad (31)$$

Fix $\theta > 0$. Then:

$$-\theta K_i + \frac{2d^2\theta^2\sigma^2}{p^2}\left(\sum_{j\in A}\tilde{X}_{ij} - \sum_{j\in B}\tilde{X}_{ij}\right)^2$$

$$= \theta\left(\sum_{j\in A}\tilde{X}_{ij} - \sum_{j\in B}\tilde{X}_{ij}\right)\left(\sum_{j\in A}\tilde{X}_{ij} + \sum_{j\in B}\tilde{X}_{ij} + 2\sum_{j\in C}\tilde{X}_{ij} - 2\frac{d}{p}\sum_{j\in S^\star}X_{ij}\right)$$

$$+ \frac{2d^2\theta^2\sigma^2}{p^2}\left(\sum_{j\in A}\tilde{X}_{ij} - \sum_{j\in B}\tilde{X}_{ij}\right)^2$$

$$= \theta\left(\sum_{j\in A}B_{ij}X_{ij} - \sum_{j\in B}B_{ij}X_{ij}\right)\times$$

$$\left(\sum_{j\in A}\left((1+\gamma(\theta))B_{ij} - \frac{2d}{p}\right)X_{ij} + \sum_{j\in B}(1-\gamma(\theta))B_{ij}X_{ij} + 2\sum_{j\in C}\left(B_{ij} - \frac{d}{p}\right)X_{ij}\right),$$

where $\gamma(\theta) = 2d^2\theta\sigma^2/p^2$. Since $\theta$ is fixed for now, we simplify the notation $\gamma(\theta)$ by simply writing $\gamma$. Let:

$$
\begin{cases}
U := \theta \left( \sum_{j \in A} B_{ij} X_{ij} - \sum_{j \in B} B_{ij} X_{ij} \right) \\
V := \sum_{j \in A} \left( (1 + \gamma) B_{ij} - \frac{2d}{p} \right) X_{ij} + \sum_{j \in B} (1 - \gamma) B_{ij} X_{ij} + 2 \sum_{j \in C} \left( B_{ij} - \frac{d}{p} \right) X_{ij}
\end{cases},
$$

so that:

$$
-\theta K_i + \frac{2d^2 \theta^2 \sigma^2}{p^2} \left( \sum_{j \in A} \tilde{X}_{ij} - \sum_{j \in B} \tilde{X}_{ij} \right)^2 = UV. \tag{32}
$$

Plugging (32) in (31), we obtain:

$$
M_{-\Delta_i}(\theta) = \mathbb{E}_{B_i} \left[ \mathbb{E}_{X_i} \left[ e^{UV} \mid B_i \right] \right]. \tag{33}
$$

Note that, conditionally on $B_i$, we have:

$$
U \overset{d}{=} \mathcal{N} \left( 0, \theta^2 \sum_{j \in A \cup B} B_{ij}^2 \right), \tag{34}
$$

and:

$$
V \overset{d}{=} \mathcal{N} \left( 0, \sum_{j \in A} \left( (1 + \gamma) B_{ij} - \frac{2d}{p} \right)^2 + \sum_{j \in B} (1 - \gamma)^2 B_{ij}^2 + 4 \sum_{j \in C} \left( B_{ij} - \frac{d}{p} \right)^2 \right). \tag{35}
$$

Let:

$$
\begin{cases}
\sigma_U^2 := \theta^2 \sum_{j \in A \cup B} B_{ij}^2 \\
\sigma_V^2 := \sum_{j \in A} \left( (1 + \gamma) B_{ij} - \frac{2d}{p} \right)^2 + \sum_{j \in B} (1 - \gamma)^2 B_{ij}^2 + 4 \sum_{j \in C} \left( B_{ij} - \frac{d}{p} \right)^2
\end{cases}.
$$

Note that:

$$
\sigma_U^2 = \theta^2 \sum_{j \in A \cup B} B_{ij},
$$

and

$$
\begin{aligned}
\sigma_V^2 &= \sum_{j \in A} \left( (1 + \gamma) B_{ij} - \frac{2d}{p} \right)^2 + \sum_{j \in B} (1 - \gamma)^2 B_{ij}^2 + 4 \sum_{j \in C} \left( B_{ij} - \frac{d}{p} \right)^2 \\
&= \sum_{j \in A} \left( (1 + \gamma)^2 B_{ij}^2 - \frac{4d(1 + \gamma)}{p} B_{ij} \right) + \frac{4d^2}{p^2} M + \sum_{j \in B} (1 - \gamma)^2 B_{ij}^2 \\
&\quad + 4 \sum_{j \in C} \left( B_{ij}^2 - \frac{2d}{p} B_{ij} \right) + \frac{4d^2}{p^2} (s - M) \\
&= \frac{4d^2 s}{p^2} + \sum_{j \in A} \left( (1 + \gamma)^2 - \frac{4d(1 + \gamma)}{p} \right) B_{ij} + \sum_{j \in B} (1 - \gamma)^2 B_{ij} + 4 \sum_{j \in C} \left( 1 - \frac{2d}{p} \right) B_{ij} \\
&= \frac{4d^2 s}{p^2} + \left( (1 + \gamma)^2 - \frac{4d(1 + \gamma)}{p} \right) \sum_{j \in A} B_{ij} + (1 - \gamma)^2 \sum_{j \in B} B_{ij} + 4 \left( 1 - \frac{2d}{p} \right) \sum_{j \in C} B_{ij}.
\end{aligned}
$$

In addition, we have:

$$
\begin{aligned}
\mathrm{Cov}(U, V) &= \theta \left( \sum_{j \in A} \left( (1 + \gamma) B_{ij} - \frac{2d}{p} \right) B_{ij} - \sum_{j \in B} (1 - \gamma) B_{ij}^2 \right) \\
&= \theta \left( \left( 1 + \gamma - \frac{2d}{p} \right) \sum_{j \in A} B_{ij} - (1 - \gamma) \sum_{j \in B} B_{ij} \right).
\end{aligned}
$$

**Lemma B.1.** *Let* $N_1 \stackrel{d}{=} \mathcal{N}\left(0, \sigma_1^2\right)$, $N_2 \stackrel{d}{=} \mathcal{N}\left(0, \sigma_2^2\right)$ *and* $\rho := corr\left(N_1, N_2\right)$. *Then:*

$$\mathbb{E}\left[e^{N_1 N_2}\right] = \begin{cases} \dfrac{1}{\sigma_1 \sigma_2 \sqrt{\left(\frac{1}{\sigma_1 \sigma_2} - \rho\right)^2 - 1}} & \text{if } \frac{1}{\sigma_1 \sigma_2} - \rho > 1, \\ +\infty & \text{else.} \end{cases}$$

*Proof.* See section B.1.1. □

Now using Lemma B.1 with Gaussian random variables $U$ and $V$, we conclude that:

$$\mathbb{E}_{X_i}\left[e^{UV} \mid B_i\right] = f\left(\theta, B_i\right), \tag{36}$$

with $f\left(\cdot, \cdot\right)$ defined in (26). Plugging this into (33), we obtain:

$$M_{-\Delta_i}\left(\theta\right) = \mathbb{E}_{B_i}\left[f\left(\theta, B_i\right)\right]. \tag{37}$$

Let $B \in \{0, 1\}^p$ be a random vector such that $B_j \stackrel{\text{i.i.d.}}{=} \text{Ber}\left(d/p\right)$. Note that $B \stackrel{d}{=} B_i$. Then (37) yields:

$$M_{-\Delta_i}\left(\theta\right) = \mathbb{E}_B\left[f\left(\theta, B\right)\right].$$

Plugging this into the Chernoff bound (30), we conclude:

$$\mathbb{P}\left(\Delta \leq 0\right) \leq \left(\inf_{\theta > 0} \mathbb{E}_B\left[f\left(\theta, B\right)\right]\right)^n.$$

□

### B.1.1 Proof of Lemma B.1

*Proof of Lemma B.1.* We have:

$\mathbb{E}\left[e^{N_1 N_2}\right]$

$$= \int_{x \in \mathbb{R}} \int_{y \in \mathbb{R}} \frac{e^{xy}}{2\pi \sigma_1 \sigma_2 \sqrt{1 - \rho^2}} \exp\left(-\frac{1}{2\left(1 - \rho^2\right)}\left(\left(\frac{x}{\sigma_1}\right)^2 - 2\rho\left(\frac{x}{\sigma_1}\right)\left(\frac{y}{\sigma_2}\right) + \left(\frac{y}{\sigma_2}\right)^2\right)\right) dydx$$

$$= \int_{x \in \mathbb{R}} \int_{y \in \mathbb{R}} \frac{1}{2\pi \sigma_1 \sigma_2 \sqrt{1 - \rho^2}} \exp\left(xy - \frac{1}{2\left(1 - \rho^2\right)}\left(\left(\frac{x}{\sigma_1}\right)^2 - 2\rho\left(\frac{x}{\sigma_1}\right)\left(\frac{y}{\sigma_2}\right) + \left(\frac{y}{\sigma_2}\right)^2\right)\right) dydx$$

$$= \int_{x \in \mathbb{R}} \int_{y \in \mathbb{R}} \frac{1}{2\pi \sqrt{1 - \rho^2}} \exp\left(\sigma_1 \sigma_2 xy - \frac{x^2 - 2\rho xy + y^2}{2\left(1 - \rho^2\right)}\right) dydx$$

$$= \int_{x \in \mathbb{R}} \int_{y \in \mathbb{R}} \frac{2\left(1 - \rho^2\right)}{2\pi \sqrt{1 - \rho^2}} \exp\left(2\sigma_1 \sigma_2\left(1 - \rho^2\right) xy - x^2 + 2\rho xy - y^2\right) dydx$$

$$= \int_{x \in \mathbb{R}} \int_{y \in \mathbb{R}} \frac{\sqrt{1 - \rho^2}}{\pi} \exp\left(2\sigma_1 \sigma_2\left(1 - \rho^2\right) xy - x^2 + 2\rho xy - y^2\right) dydx$$

Therefore:

$$\mathbb{E}\left[e^{N_1 N_2}\right] = \frac{\sqrt{1 - \rho^2}}{\pi} \int_{x \in \mathbb{R}} \int_{y \in \mathbb{R}} e^{2\sigma_1 \sigma_2\left(1 - \rho^2\right) xy - x^2 + 2\rho xy - y^2} dydx$$

$$= \frac{\sqrt{1 - \rho^2}}{\pi} \int_{x \in \mathbb{R}} \int_{y \in \mathbb{R}} e^{-\left(y - x\left(\sigma_1 \sigma_2\left(1 - \rho^2\right) + \rho\right)\right)^2 + x^2\left(\left(\sigma_1 \sigma_2\left(1 - \rho^2\right) + \rho\right)^2 - 1\right)} dydx$$

$$= \frac{\sqrt{1 - \rho^2}}{\pi} \int_{x \in \mathbb{R}} e^{x^2\left(\left(\sigma_1 \sigma_2\left(1 - \rho^2\right) + \rho\right)^2 - 1\right)} \int_{y \in \mathbb{R}} e^{-\left(y - x\left(\sigma_1 \sigma_2\left(1 - \rho^2\right) + \rho\right)\right)^2} dydx$$

$$= \frac{\sqrt{1 - \rho^2}}{\pi} \int_{x \in \mathbb{R}} e^{x^2\left(\left(\sigma_1 \sigma_2\left(1 - \rho^2\right) + \rho\right)^2 - 1\right)} \int_{y \in \mathbb{R}} \sqrt{\pi}\, \Phi_{\mathcal{N}\left(x\left(\sigma_1 \sigma_2\left(1 - \rho^2\right) + \rho\right), 1/\sqrt{2}\right)}(y)\, dydx$$

$$= \frac{\sqrt{1 - \rho^2}}{\pi} \int_{x \in \mathbb{R}} \exp\left(x^2\left(\left(\sigma_1 \sigma_2\left(1 - \rho^2\right) + \rho\right)^2 - 1\right)\right) \sqrt{\pi} dx.$$

Hence we obtain:

$$\mathbb{E}\left[e^{N_1 N_2}\right] = \frac{\sqrt{1-\rho^2}}{\sqrt{\pi}} \int_{x \in \mathbb{R}} \exp\left(x^2 \left(\left(\sigma_1 \sigma_2 \left(1 - \rho^2\right) + \rho\right)^2 - 1\right)\right) dx.$$

Let $\xi := \sigma_1 \sigma_2 \left(1 - \rho^2\right) + \rho$. Note that, if $\xi \geq 1$, the above explodes to $+\infty$. However, if $\xi < 1$, the above yields:

$$\begin{aligned}
\mathbb{E}\left[e^{N_1 N_2}\right] &= \frac{\sqrt{1-\rho^2}}{\sqrt{\pi}} \int_{x \in \mathbb{R}} \exp\left(x^2 \left(\xi^2 - 1\right)\right) dx \\
&= \frac{\sqrt{1-\rho^2}}{\sqrt{\pi}\sqrt{1-\xi^2}} \int_{x \in \mathbb{R}} \exp\left(-x^2\right) dx \\
&= \frac{\sqrt{1-\rho^2}}{\sqrt{\pi}\sqrt{1-\xi^2}} \int_{x \in \mathbb{R}} \sqrt{\pi}\, \Phi_{\mathcal{N}\left(0, 1/\sqrt{2}\right)}(x)\, dx \\
&= \frac{\sqrt{1-\rho^2}}{\sqrt{1-\xi^2}}.
\end{aligned}$$

Plugging the expression of $\xi$ in the above, we obtain:

$$\begin{aligned}
\mathbb{E}\left[e^{N_1 N_2}\right] &= \frac{\sqrt{1-\rho^2}}{\sqrt{1 - \rho^2 - \sigma_1^2 \sigma_2^2 \left(1 - \rho^2\right)^2 - 2\rho\left(1 - \rho^2\right)\sigma_1 \sigma_2}} \\
&= \frac{1}{\sqrt{1 - \sigma_1^2 \sigma_2^2 \left(1 - \rho^2\right) - 2\rho\sigma_1 \sigma_2}} \\
&= \frac{1}{\sqrt{1 + \rho^2 \sigma_1^2 \sigma_2^2 - 2\rho\sigma_1 \sigma_2 - \sigma_1^2 \sigma_2^2}} \\
&= \frac{1}{\sqrt{\left(\rho\sigma_1 \sigma_2 - 1\right)^2 - \sigma_1^2 \sigma_2^2}} \\
&= \frac{1}{\sigma_1 \sigma_2 \sqrt{\left(\frac{1}{\sigma_1 \sigma_2} - \rho\right)^2 - 1}}.
\end{aligned}$$

In addition, note that:

$$\begin{aligned}
\xi < 1 &\iff \sigma_1 \sigma_2 \left(1 - \rho^2\right) + \rho < 1 \\
&\iff \sigma_1 \sigma_2 \left(1 + \rho\right)\left(1 - \rho\right) < 1 - \rho \\
&\iff \sigma_1 \sigma_2 \left(1 + \rho\right) < 1 \\
&\iff 1 < \frac{1}{\sigma_1 \sigma_2} - \rho.
\end{aligned}$$

$\square$

## B.2 Proof of Proposition B.2

*Proof of Proposition B.2.* Fix $\eta \in (0, 1]$. To simplify notations, we will write $C^\star$ instead of $C^\star(\eta)$. All asymptotic statements are as $p \to +\infty$.

We start by showing (*i*). First, we note the following.

**Lemma B.2.**

$$\sum_{j \in A(S)} B_j = \eta \alpha \psi p \left(1 + o\left(1\right)\right), \quad \sum_{j \in B(S)} B_j = \eta \alpha \psi p \left(1 + o\left(1\right)\right), \quad \sum_{j \in C(S)} B_j = \left(1 - \eta\right) \alpha \psi p \left(1 + o\left(1\right)\right).$$

*Proof.* See section B.2.1. $\square$

Next, plugging the expressions given by Lemma B.2 into the expressions of $\sigma_U^2(\theta, B)$, $\sigma_V^2(\theta, B)$ and $\text{Cov}_{(U,V)}(\theta, B)$ given in the statement of Proposition B.1, we obtain the following.

**Proposition B.3.** *We have:*

$$\begin{cases} \sigma_U^2(\theta, B) = 2\eta\alpha\psi p\theta^2 (1 + o(1)) \\ \sigma_V^2(\theta, B) = 2\alpha\psi p \left[2 - 2\psi + \eta \left((\psi - \gamma(\theta))^2 - (1 - \psi)^2\right)\right](1 + o(1)) \\ \text{Cov}_{(U,V)}(\theta, B) = 2\eta\alpha\psi p\theta(\gamma(\theta) - \psi)(1 + o(1)) \end{cases} \quad .$$

*Proof.* See section B.2.2. □

Plugging $\xi(p)$ in the expression of $\gamma(\cdot)$ given by (27), we obtain:

$$\gamma(\xi(p)) = 2\sigma^2\psi^2\xi(p) = \frac{2\sigma^2\psi^2 C^\star}{2\alpha\psi p} = \frac{\sigma^2\psi C^\star}{\alpha p} = o(1).$$

Therefore, we have by Proposition B.3:

$$\begin{aligned} \sigma_U^2(\xi(p), B)\,\sigma_V^2(\xi(p), B) &= 4\eta\alpha^2\psi^2 p^2 \left(\frac{C^\star}{2\alpha\psi p}\right)^2 \left(2 - 2\psi + \eta\left((\psi - \gamma(\xi(p)))^2 - (1 - \psi)^2\right)\right)(1 + o(1)) \\ &= \eta C^{\star 2}\left(2 - 2\psi + \eta\left((\psi - o(1))^2 - (1 - \psi)^2\right)\right)(1 + o(1)) \\ &= \eta C^{\star 2}\left(2 - 2\psi + \eta(2\psi - 1)\right)(1 + o(1)) \\ &\xrightarrow{\text{a.s.}} \eta C^{\star 2}\left(2 - 2\psi + 2\eta\psi - \eta\right). \end{aligned}$$

Note that the above is $> 0$ because $\eta(1 - 2\psi) < 2(1 - \psi)$, which follows from the facts that $\eta \leq 1$ and $\psi \in (0, 1)$. In addition, by Proposition B.3:

$$\begin{aligned} \text{Cov}_{(U,V)}(\xi(p), B) &= 2\eta\alpha\psi p\left(\frac{C^\star}{2\alpha\psi p}\right)(\gamma(\xi(p)) - \psi)(1 + o(1)) \\ &= \eta C^\star(o(1) - \psi)(1 + o(1)) \\ &\xrightarrow{\text{a.s.}} -\eta\psi C^\star. \end{aligned}$$

Bringing these together, we obtain:

$$\begin{aligned} &\lim_{p \to +\infty} \left\{\left(1 - \text{Cov}_{(U,V)}(\xi(p), B)\right)^2 - \left(\sigma_U^2(\xi(p), B)\,\sigma_V^2(\xi(p), B) + 1 + \eta\psi C^\star\right)\right\} \\ &= \left(1 + \eta\psi C^\star\right)^2 - \eta\left(2 - 2\psi + 2\eta\psi - \eta\right)C^{\star 2} - \eta\psi C^\star - 1 \\ &= \eta^2\psi^2 C^{\star 2} + 2\eta\psi C^\star + 1 - \eta\left(2 - 2\psi + 2\eta\psi - \eta\right)C^{\star 2} - \eta\psi C^\star - 1 \\ &= \left(\eta^2\psi^2 - 2\eta + 2\eta\psi - 2\eta^2\psi + \eta^2\right)C^{\star 2} + \eta\psi C^\star \\ &= \eta C^\star\left(-(1 - \psi)\left(2 - \eta(1 - \psi)\right)C^\star + \psi\right) \\ &= 0. \end{aligned}$$

In addition:

$$\lim_{p \to +\infty} \left\{ \frac{1}{\sigma_U\left(\xi\left(p\right), B\right) \sigma_V\left(\xi\left(p\right), B\right)} - \rho\left(\xi\left(p\right), B\right) \right\}^2$$

$$= \lim_{p \to +\infty} \left\{ \frac{1 - \mathrm{Cov}_{(U,V)}\left(\xi\left(p\right), B\right)}{\sigma_U\left(\xi\left(p\right), B\right) \sigma_V\left(\xi\left(p\right), B\right)} \right\}^2$$

$$= \frac{\left(1 + \eta\psi C^\star\right)^2}{\eta\left(2 - 2\psi + 2\eta\psi - \eta\right) C^{\star 2}}$$

$$= 1 + \frac{\left(1 + \eta\psi C^\star\right)^2 - \eta\left(2 - 2\psi + 2\eta\psi - \eta\right) C^{\star 2}}{\eta\left(2 - 2\psi + 2\eta\psi - \eta\right) C^{\star 2}}$$

$$= 1 + \frac{1 + \eta\psi C^\star}{\left(1 + \eta\psi C^\star\right)^2 - \left(1 + \eta\psi C^\star\right)}$$

$$= 1 + \frac{1}{\eta\psi C^\star} > 1.$$

Therefore, there exists $p' \in \mathbb{N}$ such that, for all $p \geq p'$ we have:

$$\left\{ \frac{1}{\sigma_U\left(\xi\left(p\right), B\right) \sigma_V\left(\xi\left(p\right), B\right)} - \rho\left(\xi\left(p\right), B\right) \right\}^2 > 1.$$

Then note that, for all $p \geq p'$ we have:

$$\frac{1}{\sigma_U\left(\xi\left(p\right), B\right) \sigma_V\left(\xi\left(p\right), B\right)} - \rho\left(\xi\left(p\right), B\right) > 1,$$

and hence:

$$f\left(\xi\left(p\right), B\right) = \frac{1}{\sigma_U\left(\xi\left(p\right), B\right) \sigma_V\left(\xi\left(p\right), B\right) \sqrt{\left(\frac{1}{\sigma_U(\xi(p),B)\sigma_V(\xi(p),B)} - \rho\left(\xi\left(p\right), B\right)\right)^2 - 1}}$$

$$= \frac{1}{\sqrt{\left(1 - \mathrm{Cov}_{(U,V)}\left(\xi\left(p\right), B\right)\right)^2 - \sigma_U^2\left(\xi\left(p\right), B\right) \sigma_V^2\left(\xi\left(p\right), B\right)}}$$

$$\xrightarrow{\text{a.s.}} \sqrt{\frac{1}{1 + \eta\psi C^\star}}.$$

Therefore, we conclude that (*i*) holds. We now show (*ii*). We have:

$$\inf_{\theta > 0} \mathbb{E}_B\left[f\left(\theta, B\right)\right] \leq \mathbb{E}_B\left[f\left(\xi\left(p\right), B\right)\right] = \mathbb{E}_B\left[H_p\left(B\right)\right].$$

Plugging this into the Chernoff bound obtained in Proposition B.1, we obtain:

$$\mathbb{P}\left(\Delta \leq 0\right) \leq \left(\inf_{\theta > 0} \mathbb{E}_B\left[f\left(\theta, B\right)\right]\right)^n \leq \left(\mathbb{E}_B\left[H_p\left(B\right)\right]\right)^n.$$

$\square$

### B.2.1  Proof of Lemma B.2

*Proof of Lemma B.2.* We know that $B_j \overset{\text{i.i.d.}}{\sim} \mathrm{Ber}\left(\psi\right)$. Therefore:

$$\sum_{j \in A(S)} B_j = |A\left(S\right)|\psi + \sqrt{|A\left(S\right)|\psi} N_A + o\left(\sqrt{|A\left(S\right)|\psi}\right),$$

where $N_A \sim \mathcal{N}\left(0, 1\right)$. In addition, we have:

$$|A\left(S\right)| = \lceil \eta s \rceil = \eta s\left(1 + o\left(1\right)\right).$$

Hence:

$$\sum_{j \in A(S)} B_j = \eta \alpha \psi p \big(1 + o\,(1)\big).$$

Similarly, we have:

$$\sum_{j \in B(S)} B_j = \eta \alpha \psi p \big(1 + o\,(1)\big) \quad \text{and} \quad \sum_{j \in C(S)} B_j = (1 - \eta)\,\alpha \psi p \big(1 + o\,(1)\big).$$

$\square$

### B.2.2 Proof of Proposition B.3

*Proof of Proposition B.3.* We have by Proposition B.1:

$$\sigma_U^2\,(\theta, B) = \theta^2 \sum_{j \in A(S) \cup B(S)} B_j,$$

$$\sigma_V^2\,(\theta, B) = \frac{4d^2 s}{p^2} + \left( \big(1 + \gamma\,(\theta)\big)^2 - \frac{4d\big(1 + \gamma\,(\theta)\big)}{p} \right) \sum_{j \in A(S)} B_j$$
$$+ \big(1 - \gamma\,(\theta)\big)^2 \sum_{j \in B(S)} B_{ij} + 4\left(1 - \frac{2d}{p}\right) \sum_{j \in C(S)} B_j,$$

$$\mathrm{Cov}_{(U,V)}\,(\theta, B) = \theta \left( \left(1 + \gamma\,(\theta) - \frac{2d}{p}\right) \sum_{j \in A(S)} B_{ij} - \big(1 - \gamma\,(\theta)\big) \sum_{j \in B(S)} B_{ij} \right).$$

Now by Lemma B.2 we have:

$$\begin{cases} \sum_{j \in A(S)} B_j = \eta \alpha \psi p \big(1 + o\,(1)\big) \\ \sum_{j \in B(S)} B_j = \eta \alpha \psi p \big(1 + o\,(1)\big) \\ \sum_{j \in C(S)} B_j = (1 - \eta)\,\alpha \psi p \big(1 + o\,(1)\big) \end{cases}.$$

Hence, the expression of $\sigma_U^2\,(\theta, B)$ writes:

$$\begin{aligned} \sigma_U^2\,(\theta, B) &= \theta^2 \sum_{j \in A(S)} B_j + \theta^2 \sum_{j \in B(S)} B_j \\ &= \theta^2 \eta \alpha \psi p \big(1 + o\,(1)\big) + \theta^2 \eta \alpha \psi p \big(1 + o\,(1)\big) \\ &= 2 \eta \alpha \psi p \theta^2 \big(1 + o\,(1)\big). \end{aligned}$$

The expression of $\sigma_V^2(\theta, B)$ writes:

$$\sigma_V^2(\theta, B)$$

$$= \frac{4d^2 s}{p^2} + \left( \left(1 + \gamma(\theta)\right)^2 - \frac{4d\left(1 + \gamma(\theta)\right)}{p} \right) \sum_{j \in A(S)} B_j$$

$$+ \left(1 - \gamma(\theta)\right)^2 \sum_{j \in B(S)} B_{ij} + 4\left(1 - \frac{2d}{p}\right) \sum_{j \in C(S)} B_j$$

$$= 4\alpha\psi^2 p + \left(1 + \gamma(\theta)\right)\left(1 + \gamma(\theta) - 4\psi\right)\eta\alpha\psi p\left(1 + o(1)\right)$$

$$+ \left(1 - \gamma(\theta)\right)^2 \eta\alpha\psi p\left(1 + o(1)\right) + 4\left(1 - 2\psi\right)\left(1 - \eta\right)\alpha\psi p\left(1 + o(1)\right)$$

$$= \alpha\psi p\left(4\psi + \eta\left(1 + \gamma(\theta)\right)\left(1 + \gamma(\theta) - 4\psi\right) + \eta\left(1 - \gamma(\theta)\right)^2 + 4\left(1 - 2\psi\right)\left(1 - \eta\right)\right)\left(1 + o(1)\right)$$

$$= \alpha\psi p\left(4 - 4\psi + \eta\left(\left(1 + \gamma(\theta)\right)^2 + \left(1 - \gamma(\theta)\right)^2 - 4\psi\left(1 + \gamma(\theta)\right) - 4\left(1 - 2\psi\right)\right)\right)\left(1 + o(1)\right)$$

$$= \alpha\psi p\left(4 - 4\psi + \eta\left(2 + 2\gamma(\theta)^2 - 4\psi\gamma(\theta) + 4\psi - 4\right)\right)\left(1 + o(1)\right)$$

$$= 2\alpha\psi p\left(2 - 2\psi + \eta\left(\gamma(\theta)^2 - 2\psi\gamma(\theta) + 2\psi - 1\right)\right)\left(1 + o(1)\right)$$

$$= 2\alpha\psi p\left(2 - 2\psi + \eta\left(\gamma(\theta)^2 - 2\psi\gamma(\theta) + \psi^2 - \psi^2 + 2\psi - 1\right)\right)\left(1 + o(1)\right)$$

$$= 2\alpha\psi p\left[2 - 2\psi + \eta\left(\left(\psi - \gamma(\theta)\right)^2 - \left(1 - \psi\right)^2\right)\right]\left(1 + o(1)\right).$$

The expression of $\mathrm{Cov}_{(U,V)}(\theta, B)$ writes:

$$\mathrm{Cov}_{(U,V)}(\theta, B) = \theta\left(\left(1 + \gamma(\theta) - \frac{2d}{p}\right)\sum_{j \in A(S)} B_{ij} - \left(1 - \gamma(\theta)\right)\sum_{j \in B(S)} B_{ij}\right)$$

$$= \theta\left(\left(1 + \gamma(\theta) - 2\psi\right)\eta\alpha\psi p\left(1 + o(1)\right) - \left(1 - \gamma(\theta)\right)\eta\alpha\psi p\left(1 + o(1)\right)\right)$$

$$= \eta\alpha\psi p\theta\left(\left(1 + \gamma(\theta) - 2\psi\right) - \left(1 - \gamma(\theta)\right)\right)\left(1 + o(1)\right)$$

$$= 2\eta\alpha\psi p\theta\left(\gamma(\theta) - \psi\right)\left(1 + o(1)\right),$$

as desired. $\qquad\square$

