# OpenReview forum: "The Price of Sparsity: Sufficient Conditions for Sparse Recovery using Sparse and Sparsified Measurements"
_NeurIPS.cc/2025/Conference — NeurIPS 2025 poster_

### Official Review · Reviewer_qyQw · 2025-06-18

**Clarity:** 4
**Significance:** 4
**Originality:** 3
**Rating:** 6
**Confidence:** 4

**Summary:**

The paper studies sparse recovery with the added structure that the design matrix itself may be sparse (i.e. sparse measurements). The authors derive the information-theoretic threshold which delineates the sample complexity for when asymptotic support recovery of the signal is possible/impossible. Under a certain conjecture, the authors also describe how one might sparsify a dense measurement matrix and still obtain signal recovery; this can be useful from the standpoint of storage and computation resources.

**Questions:**

- The authors are transparent about the gaps. Though it may not be possible to quickly resolve them, I think it would substantially improve the paper if the authors could describe, even at a high-level, what the difficulties are in closing the gaps. Such a description would be helpful in convincing even the skeptical reader that the issue is not so serious (as the authors believe).

- The authors need to acknowledge and cite the missing data literature, as this is very closely related to the sparsification considered in the latter part of the paper. For example, the authors need to discuss Loh & Wainwright, "High-dimensional regression with noisy and missing data: Provable guarantees with nonconvexity" as well as discuss related references.

**Ethical Concerns:**

["NO or VERY MINOR ethics concerns only"]

**Final Justification:**

The authors have addressed the gap that I noted in my comments, and so I have increased my original score (5 to 6).

**Limitations:**

Yes

**Paper Formatting Concerns:**

No concerns

**Quality:**

3

**Strengths And Weaknesses:**

__Strengths__

The paper studies a canonical setting and the problem of studying sparse measurement matrices is very interesting. The problem is clearly of interest to the NeurIPS community, and the paper provides interesting practical implications as well. The paper's main result, a sharp characterization of the price of sparsity, is quite fundamental and nicely explains to what degree information in the measurement matrix is important. The writing is very clear, and the arguments are transparent. I especially appreciated the proof sketch of Theorem 1. The paper is also very honest about its limitations. I enjoyed reading this paper.

__Weaknesses__

In my view, the only weaknesses of the paper are the ones the authors themselves identify. Strictly speaking, the upper and lower bounds are not directly comparable since they apply to different settings. Yet I, like the authors, don't believe the gap is fundamental and could be closed. Of course, it does weaken the ultimate conclusions of the paper. Specifically, the conclusion of Theorem 1 + Corollary 2 cannot be interpreted as a strict characterization of the information-theoretic threshold. As the authors note in Remark 2.2, the parameter spaces between upper and lower bounds are different, and the notion of recovery is also different. Likewise, the results regarding sparsification are conditional on a conjecture. It would be really great to have all this cleaned up (especially concerning Theorem 1 + Corollary 2), especially for a top venue like NeurIPS. However, I am sympathetic to the authors and also am willing to speculate the gap is not fundamental.

---

> ### Author Rebuttal · Authors · 2025-07-30
>
> &#8203;
>
> We thank the reviewer for their thorough evaluation. Below, we address their questions and concerns to the best of our ability.
>
> &#8203;
>
> ---
>
> &#8203;
>
> >- **Weaknesses:** In my view, the only weaknesses of the paper are the ones the authors themselves identify. Strictly speaking, the upper and lower bounds are not directly comparable since they apply to different settings. Yet I, like the authors, don't believe the gap is fundamental and could be closed. Of course, it does weaken the ultimate conclusions of the paper. Specifically, the conclusion of Theorem 1 + Corollary 2 cannot be interpreted as a strict characterization of the information-theoretic threshold. As the authors note in Remark 2.2, the parameter spaces between upper and lower bounds are different, and the notion of recovery is also different. Likewise, the results regarding sparsification are conditional on a conjecture. It would be really great to have all this cleaned up (especially concerning Theorem 1 + Corollary 2), especially for a top venue like NeurIPS. However, I am sympathetic to the authors and also am willing to speculate the gap is not fundamental.
>
> &#8203;
>
> - **Rebuttal:** We thank the reviewer for their comment on the limitations of our work. We note the following about the limitations described in Remark 2.2.
> &#8203;
>    1. ___Matching signal classes in Corollary 2 (Remark 2.2, first bullet point).___ Revisiting the proof of the necessity statement of Wang et al. [16, Theorem 2], we verified that the same Fano-based argument applies verbatim to show the necessity statement for binary signals, i.e. $\beta^\star \in \\{ 0,1 \\}^p$. In fact, at a high level, the authors show the necessary condition for $\beta^\star \in C_{p,s}$ by first showing it for $\beta^\star \in \\{0,1\\}^p$ then adding that if no decoder can recover $\beta^\star \in \\{0,1\\}^p$ then no decoder can recover $\beta^\star \in C_{p,s}$ (since $\\{0,1\\}^p \subseteq C_{p,s}$), which weakens their result. This closes the gap between signal spaces, unifying both necessary and sufficient conditions for binary signals. In the camera‑ready version we will restate the necessary condition of Corollary 2 for binary signals and remove the first bullet point of Remark 2.2.
> &#8203;
>
>    2. ___Inconsistency of the notions of "reliable recovery" in necessity and sufficiency statements (Remark 2.2, second bullet point).___  In line 197, we mistakenly described Reeves et al. [13] as proving exact‑support recovery (i.e. equality of supports). In fact, they study a random binary signal model (uniform over $s$‑sparse supports) and assess performance via MSE relative to a random‑guess baseline. They show an all‑or‑nothing transition: for $n > (1+\varepsilon) n_{\text{INF}}$, the MLE achieves strong recovery in the sense that the ratio of its MSE to the random‑guess MSE tends to $0$; while for $n < (1-\varepsilon) n_{\text{INF}}$, no estimator can asymptotically perform better than a random guess. In particular, they show convergence of the expected rescaled symmetric difference error to $0$. By contrast, Theorem 1 in our work establishes convergence in probability of the normalized symmetric difference for each fixed signal. Thus their guarantee is expectation‑based under a random‑signal model, whereas ours is probability‑based under a fixed‑signal model. Although their proof relies on the Gaussianity of the entries of $X$, which breaks down in our sparse setting, we believe the gap between their expectation guarantee and our high‑probability guarantee can be closed in the sparse‑measurement setting with more refined tools similar to those in our proof of Theorem 1, especially since the same sequence of proofs happened in the literature of the dense setting. For the same reason, we believe a similar negative result to the one shown by Reeves et al. [13] can be shown in the sparse setting. In the camera-ready version, we will correct Remark 2.2 accordingly.
>
>    &#8203;
>
>    Finally, concerning Conjecture B1: it's about a uniform integrability statement, the proof of which we believe is a tedious task due to the complications that arise from introducing the binary mask of sparsification on one hand and from the bias in the rescaled observations (discussed in lines 279-286 of the manuscript) on the other. We believe the proof of Conjecture B.1 would be a far more technically involved sparsification-version of the proof of Lemma A.2.
>
> &#8203;
>
> ---
>
> &#8203;
>
> >- **Questions:** The authors are transparent about the gaps. Though it may not be possible to quickly resolve them, I think it would substantially improve the paper if the authors could describe, even at a high level, what the difficulties are in closing the gaps. Such a description would be helpful in convincing even the skeptical reader that the issue is not so serious (as the authors believe).
>
> &#8203;
>
> - **Rebuttal:** We thank the reviewer for this interesting question. With the above modifications of Remark 2.2 in mind, there aren’t any gaps anymore. There are two slightly different problem setups, with different error metrics and recovery guarantees. While in our problem setup the recovery guarantees we obtained using our error metric did not allow us to use the same notion of "recovery" in the necessity and sufficiency statements, Reeves et al. [13] obtain (in their problem setup and using their error metric) recovery guarantees that allow a single notion of "recovery", which yields a cleaner phase transition. As mentioned above, we believe that a result similar to theirs can be shown in the sparse setting. However, that would constitute a completely different work, which we leave for future exploration.
>
> &#8203;
>
> ---
>
> &#8203;
>
> >- **Questions:** The authors need to acknowledge and cite the missing data literature, as this is very closely related to the sparsification considered in the latter part of the paper. For example, the authors need to discuss Loh \& Wainwright, "High-dimensional regression with noisy and missing data: Provable guarantees with nonconvexity" as well as discuss related references.
>
> &#8203;
>
> - **Rebuttal:** We thank the reviewer for pointing this out. Sparsification is indeed closely related to the "missing covariates" model studied in the high‑dimensional regression literature. We will add a short discussion and cite Loh and Wainwright, "High-dimensional regression with noisy and missing data: Provable guarantees with nonconvexity", and subsequent work on noisy or missing data. Their results provide algorithmic $\ell_2$ and $\ell_1$ error bounds for dense Gaussian designs with a constant missingness rate, whereas our work focuses on information‑theoretic support‑recovery thresholds: we allow the missingness rate to grow (via $d$ or $\psi$), and quantify the exact price of sparsification. In the camera‑ready version of the paper, we will highlight how our findings complement the existing missing‑data guarantees and direct readers to these references.
>
> &#8203;

---

> ### Comment · Reviewer_qyQw · 2025-08-05
>
> Thanks very much for your response! I'm quite satisfied, especially with the modifications to Remark 2.2. Since the gaps have been closed, I will increase my score.

---

### Official Review · Reviewer_A45Y · 2025-07-01

**Clarity:** 4
**Significance:** 3
**Originality:** 3
**Rating:** 5
**Confidence:** 4

**Summary:**

The paper studied the problem of support recovery of a sparse vector (signal) of dimension p from n linear projections, under the less common but interesting assumption that each projection vector is sparse (with d non-zero elements on average). In other words, support recovery of a sparse vector using a sparse sensing matrix.
The main results are:
 (1) A sufficient condition on the number of measurements guaranteeing recovery using the MLE in terms of p, d, and s (the number of non-zero entries in the signal) when ds/p-> infinity. Combined with previous results (Wang, Wainwright, and Ramchandran, 2016), this new condition yields an information-theoretic phase transition.
 (2) Sufficient conditions on the number of samples n when a dense sensing matrix undergoes a sparsification process.

**Questions:**

Referring to an extended setting outlined in Lines 67-68, it is unclear what sparsity means.

**Ethical Concerns:**

["NO or VERY MINOR ethics concerns only"]

**Final Justification:**

The authors provided satisfactory replies to my question and addressed the weaknesses I raised.

The author's response strengthens my opinion to accept the paper.

**Limitations:**

Yes. Especially in Remark 2.2.

**Paper Formatting Concerns:**

No formatting concerns.

**Quality:**

4

**Strengths And Weaknesses:**

Strengths:
- The topic is within the scope of NeurIPS.
- The motivation for considering a sparse sensing matrix to save computation is reasonable.
- The contributions are significant.

Weaknesses:
- As the authors mentioned, the analysis in the paper leaves several challenges (mostly discussed in Remark 2.2). These challenges could probably be resolved with relatively little work, so it appears that a more complete work is within reach.
- I suggest explaining in the introduction the connection between support recovery and MSE or other natural error criteria.
- There are a few undefined terms: H(\alpha), ||s||_0, in the abstract, it is unclear that n_{inf}^{sp} refers to the number of measurements,

---

> ### Author Rebuttal · Authors · 2025-07-30
>
> &#8202;
>
> We thank the reviewer for their thorough evaluation. Below, we address their questions and concerns to the best of our ability.
>
> &#8202;
>
> ---
>
> &#8202;
>
> >- **Weaknesses:** As the authors mentioned, the analysis in the paper leaves several challenges (mostly discussed in Remark 2.2). These challenges could probably be resolved with relatively little work, so it appears that a more complete work is within reach.
>
> &#8202;
>
> - **Rebuttal:** We thank the reviewer for their comment on the limitations of our work. We note the following about the limitations described in Remark 2.2.
> &#8202;
>
>    1. ___Matching signal classes in Corollary 2 (Remark 2.2, first bullet point).___ Revisiting the proof of the necessity statement of Wang et al. [16, Theorem 2], we verified that the same Fano-based argument applies verbatim to show the necessity statement for binary signals, i.e. $\beta^\star \in \\{0,1\\}^p$. In fact, at a high level, the authors show the necessary condition for $\beta^\star \in C_{p,s}$ by first showing it for $\beta^\star \in \\{0,1\\}^p$ then adding that if no decoder can recover $\beta^\star \in \\{0,1\\}^p$ then no decoder can recover $\beta^\star \in C_{p,s}$ (since $\\{0,1\\}^p \subseteq C_{p,s}$), which weakens their result. This closes the gap between signal spaces, unifying both necessary and sufficient conditions for binary signals. In the camera‑ready version, we will restate the necessary condition of Corollary 2 for binary signals and remove the first bullet point of Remark 2.2.
>    &#8202;
>
>    2. ___Inconsistency of the notions of "reliable recovery" in necessity and sufficiency statements (Remark 2.2, second bullet point).___ In line 197, we mistakenly described Reeves et al. [13] as proving exact‑support recovery (i.e., equality of supports). In fact, they study a random binary signal model (uniform over $s$‑sparse supports) and assess the performance of a decoder via MSE relative to a random‑guess baseline. They show an all‑or‑nothing transition: for $n > (1+\varepsilon) n_{\text{INF}}$, the MLE achieves strong recovery in the sense that the ratio of its MSE to the random‑guess MSE tends to $0$; while for $n < (1-\varepsilon) n_{\text{INF}}$, no estimator can asymptotically perform better than a random guess. In particular, they show convergence of the expected rescaled symmetric‑difference error to $0$. By contrast, Theorem 1 in our work establishes convergence in probability of the normalized symmetric difference for each fixed signal. Thus, their guarantee is expectation‑based under a random‑signal model, whereas ours is probability‑based under a fixed‑signal model. Finally, although their proof relies on the Gaussianity of the entries of $X$, which breaks down in our sparse setting, we believe the gap between their expectation guarantee and our high‑probability guarantee can be closed in the sparse‑measurement setting with more refined tools similar to those in our proof of Theorem 1, especially since the same sequence of proofs happened in the literature of the dense setting. For the same reason, we believe a similar negative result to the one shown by Reeves et al. [13] can be shown in the sparse setting. In the camera-ready version, we will correct Remark 2.2 accordingly.
>    &#8202;
>
>     With the above modifications of Remark 2.2 in mind, there aren’t any challenges anymore in the first part of the paper. Our work and the work of Reeves et al. [13] constitute two slightly different problem setups, with different error metrics and recovery guarantees. While in our problem setup the recovery guarantees we obtained using our error metric did not allow us to use the same notion of "recovery" in the necessity and sufficiency statements, Reeves et al. [13] obtain (in their problem setup and using their error metric) recovery guarantees that allow a single notion of "recovery", which yields a cleaner phase transition. As mentioned above, we believe that a result similar to theirs can be shown in the sparse setting. However, that would constitute a completely different work, which we leave for future exploration.
>
>    &#8202;
>
>    Finally, concerning Conjecture B1: it's about a uniform integrability statement, the proof of which we believe is a tedious task due to the complications that arise from introducing the binary mask of sparsification on one hand and from the bias in the rescaled observations (discussed in lines 279-286 of the manuscript) on the other. We believe the proof of Conjecture B.1 would be a far more technically involved sparsification-version of the proof of Lemma A.2.
>
> &#8202;
>
> ---
>
> &#8202;
>
> >- **Weaknesses:** I suggest explaining in the introduction the connection between support recovery and MSE or other natural error criteria.
>
> &#8202;
>
> - **Rebuttal:** We thank the reviewer for their helpful suggestion. Since the signal is binary with fixed sparsity, our error metric (the symmetric difference; see Definition 2.3) corresponds exactly to the squared $\ell_2$-loss. Specifically, for any $s$-sparse vectors $\beta, \beta^\star \in \\{0,1\\}^p$, we have $ \mathrm{|} \\ \mathrm{Supp}(\beta) \\ \Delta \\ \mathrm{Supp}(\beta^\star) \\ \mathrm{|}  = \mathrm{||}\beta - \beta^\star \mathrm{||}\_2^2$. Thus, Theorem 1 can also be interpreted as a high-probability guarantee on the convergence of the rescaled squared loss (per non-zero coordinate).
>     In our work, we focus on binary signals as a simplifying assumption. While this setting is representative, sampling complexity guarantees for support recovery in this class can typically be extended to the class of signals whose non-zero components are bounded away from $0$ by $1$, i.e. $\min_{j = 1,\dots,p} |\beta_j^\star| \geq 1$, with the same thresholds.
>     Moreover, sparse linear regression can naturally be reduced to support recovery since, once the support is identified, the full signal can be estimated via the closed-form expression of the maximum-likelihood estimator given by $\beta_{\text{MLE}} = X_{S^\star}^{+} Y$, where $X_{S^\star}^{+}$ denotes the Moore-Penrose pseudoinverse of the submatrix of $X$ restricted to the support $S^\star = \text{Supp}(\beta^\star)$. We agree with the reviewer that these clarifications are valuable and will include them in the introduction of the camera-ready version of the paper.
>
> &#8202;
>
> ---
>
> &#8202;
>
> >- **Weaknesses:** There are a few undefined terms: $H(\alpha)$, $\mathrm{||} s \mathrm{||} \_0$, in the abstract, it is unclear that $n_{\text{INF}}^{\text{SP}}$ refers to the number of measurements.
>
> &#8202;
>
> - **Rebuttal:** We agree with the reviewer on this weakness and thank them for pointing it out. In the camera-ready version of the paper, we will:
>     * Add to the abstract a clarification that $n_{\text{INF}}^{\text{SP}}$ refers to the number of measurements.
>     * Add a “Notation” paragraph at the end of the Introduction (Section 1) to clarify some notations like the entropy function $H(\cdot)$ and the $\ell_0$-norm $\mathrm{||} \cdot \mathrm{||}\_0$.
>
> &#8202;
>
> ---
>
> &#8202;
>
> >- **Questions:** Referring to an extended setting outlined in Lines 67-68, it is unclear what sparsity means.
>
> &#8202;
>
> - **Rebuttal:** We thank the reviewer for asking for clarification. In the extended setting (lines 67-68), "sparsity" continues to mean having exactly $s$ non-zero components, i.e. $\mathrm{||}\beta^\star\mathrm{||}\_0 = s$. The only change is that we now allow each non-zero coordinate to take an arbitrary real value of magnitude at least $\lambda$, replacing the binary constraint $\beta^\star \in \\{0,1\\}^p$ by $\beta^\star \in \\{ \\ \beta \in \mathbb{R}^p \\ \colon \\ \min_{j \in [p]} |\beta_j| \geq \lambda \\ \\}$. Equivalently, $\beta^\star \in \\{ \\ \beta \in \mathbb{R}^p \\ \colon \\ \mathrm{||}\beta\mathrm{||}\_0 = s, \\ \min_{j \in [p]} |\beta_j| \geq \lambda \\ \\}$. We will add this explicit statement to the text in the camera-ready version to remove any ambiguity.
>
> &#8202;

---

> ### Comment · Reviewer_A45Y · 2025-07-31
>
> Thank you for clarifying the points I raised. My opinion of the paper remains very positive and I have no further questions.

---

### Official Review · Reviewer_KJwa · 2025-07-03

**Clarity:** 4
**Significance:** 3
**Originality:** 3
**Rating:** 5
**Confidence:** 4

**Summary:**

The authors study how taking a sparse measurement matrix in sparse regression (or compressed sensing) problems affects the sample complexity of these problems. Previous work showed that, at a given level of sparsity, recovery of the hidden signal is impossible with too few statements. This work shows a matching positive information-theoretic result, that above this number of samples the (computationally intractable) maximum likelihood estimator does asymptotically recover the (support of the) signal. Thus this result settles the information-theoretic threshold of sparse regression with a sparse measurement matrix. To account for the possibility that the measurement matrix is prescribed rather than designed by the user, the authors also prove a similar result (conditional on a technical conjecture) for a random sparsification of a given dense measurement matrix. That is, a measurement is *generated* with a dense measurement matrix, but *recovered* by a sparsification of that matrix.

**Questions:**

- Since the idea of sparsification is an algorithmic one and the precise means of sparsification is presumably flexible, could other ways of sparsifying a matrix rather than completely at random that have stronger guarantees on spectral properties they preserve be useful? This area has been widely studied with the advent of general-purpose matrix concentration inequalities, and for example many examples are discussed in Tropp's monograph "An introduction to matrix concentration inequalities".

- What does "slow decay of sparsity" mean in regard to the computational threshold (p2), and could the authors be slightly more specific about the status of the algorithmic threshold? Is the algorithmic threshold for a sparse measurement matrix actually known, and known to be comparable to the performance of LASSO? I assume not quite, since (to my knowledge) the question of the overlap gap property under sparsity of the measurement matrix has not been addressed before?

**Ethical Concerns:**

["NO or VERY MINOR ethics concerns only"]

**Final Justification:**

I did not raise any substantial issues but just a few clarifying questions, and the authors replied to my satisfaction. I am leaving my positive rating of the paper unchanged.

**Limitations:**

Yes.

**Paper Formatting Concerns:**

N/A.

**Quality:**

4

**Strengths And Weaknesses:**

Strengths: The main result is well-motivated and a valuable addition to the literature on this subject that closes a clear open problem of pinning down the information-theoretic threshold for this problem. The authors clearly describe the state of affairs and their contribution, and the background information on the problem is presented very nicely and leaves (to me) no lingering concerns. The idea of sparsification is also a nice one and gives a valuable complement to the first main result about sparse measurement matrices. The authors are also very clear and explicit about even minor limitations and restrictions of their result, which I appreciated.

Weaknesses: The proof techniques for analyzing the MLE are fairly standard, and it seems (though I have not gone through them in detail) that while sparsity changes things it does not make these calculations require fundamentally different tools. Also, a minor point: the introduction of sparsification is a little bit unclear, and I would present this problem in the way that I describe it in the summary, emphasizing that the point is to consider using a different matrix for measuring versus recovering.

---

> ### Author Rebuttal · Authors · 2025-07-30
>
> &#8202;
>
> We thank the reviewer for their thorough evaluation. Below, we address their questions and concerns to the best of our ability. At the end, we note a slight improvement to our work that we will incorporate into the camera‑ready version of the paper.
>
> &#8202;
>
> ---
>
> &#8202;
>
> >- **Weaknesses:** The proof techniques for analyzing the MLE are fairly standard, and it seems (though I have not gone through them in detail) that while sparsity changes things it does not make these calculations require fundamentally different tools.
>
> &#8202;
>
> - **Rebuttal:** We appreciate the reviewer’s observation. While it is true that our proofs rely on classical tools such as the union bound and the moment method, the sparse setting introduces new technical challenges that go beyond those in the dense case. In particular, the sparsity of the design matrix breaks the Gaussian structure typically leveraged in the dense setting, requiring us to analyze more intricate moment generating functions. Most importantly, this involves studying transformations of Binomial random variables with means diverging to infinity, as seen in the proof of Proposition A.1, as well as Lemma A.1 and Lemma A.2. These challenges become even more pronounced in the second part of the paper (Section 3), particularly in the proofs of Proposition B.2 and Theorem 3 where the result remains subject to a subtle conjecture (Conjecture B.1) due to technical complications that arise from the Bernoulli mask used in sparsification. We believe that handling such complications is a meaningful technical contribution, and we thank the reviewer for highlighting the similarity to the dense setting in the use of traditional techniques.
>
> &#8202;
>
> ---
>
> &#8202;
>
> >- **Weaknesses:** Also, a minor point: the introduction of sparsification is a little bit unclear, and I would present this problem in the way that I describe it in the summary, emphasizing that the point is to consider using a different matrix for measuring versus recovering.
>
> &#8202;
>
> - **Rebuttal:** We thank the reviewer for the phrasing suggestion. We agree that framing the sparsification setting as using a different matrix for measurement versus recovery provides a  clearer perspective. We will incorporate this description into the camera-ready version of the paper to improve clarity.
>
> &#8202;
>
> ---
>
> &#8202;
>
> >- **Questions:** Since the idea of sparsification is an algorithmic one and the precise means of sparsification is presumably flexible, could other ways of sparsifying a matrix rather than completely at random that have stronger guarantees on spectral properties they preserve be useful? This area has been widely studied with the advent of general-purpose matrix concentration inequalities, and for example many examples are discussed in Tropp's monograph "An introduction to matrix concentration inequalities".
>
> &#8202;
>
> - **Rebuttal:** We thank the reviewer for these insightful suggestions. We view our work as a first exploratory step toward understanding the role of sparsification in signal recovery, rather than an attempt to close the question. In this paper, we deliberately focus on a simple and arguably naïve sparsification model (applying an i.i.d. Bernoulli mask after the dense matrix is observed) to introduce the question and highlight its fundamental challenges. This choice keeps the setting directly comparable (and technically equivalent) to the classical missing‑at‑random framework studied extensively in high‑dimensional statistics, see Loh and Wainwright "High-dimensional regression with noisy and missing data: Provable guarantees with nonconvexity" and related references. In the camera-ready version of the paper, we will highlight how our findings complement the existing missing-data guarantees and direct readers to these references.
>    In general, the designer-dependent sparsification is an appropriate setting for signal processing, but less appropriate in statistical applications. From a pure sparsification standpoint, we agree that more sophisticated sparsification schemes -- particularly those designed to preserve spectral properties -- are a promising direction for sharpening recovery guarantees and improving empirical performance. We thank the reviewer for these great suggestions and believe they constitute an excellent direction for future exploration of the sparsification problem.
>
> &#8202;
>
> ---
>
> &#8202;
>
> >- **Questions:** What does "slow decay of sparsity" mean in regard to the computational threshold (p2), and could the authors be slightly more specific about the status of the algorithmic threshold? Is the algorithmic threshold for a sparse measurement matrix actually known, and known to be comparable to the performance of LASSO? I assume not quite, since (to my knowledge) the question of the overlap gap property under sparsity of the measurement matrix has not been addressed before?
>
> &#8202;
>
> - **Rebuttal:** We thank the reviewer for their interesting question. By the “slow decay of sparsity” in line 82, we mean the assumption made by Omidiran and Wainwright [12]. This assumption is later stated in the equation following line 87, and can also be found in [12], equation (8c). Under this assumption, the authors show in [12, Theorem 1] that the sufficient condition under which the LASSO reliably recovers the signal in the dense case, notably: $n > (1+\varepsilon) n_{\text{ALG}}$, where $n_{\text{ALG}} = 2 s \log(p-s)$ is also sufficient, in the sparse case, for the LASSO to reliably recover the (signed) support of the signal. We believe this is interesting since it might suggest that the algorithmic threshold is insensitive to the sparsity of the design although, as far as we know, and as the reviewer mentioned, there are yet no negative results on the algorithmic threshold side, since the overlap gap property has not been addressed before in the sparse measurement matrix setting. To summarize, here is the current status of the algorithmic threshold, to the best of our knowledge:
>
>    * ___Positive results:___ only known so far is by Omidiran and Wainwright [12, Theorem 1].
>    * ___Negative results:___ none yet.
>
> &#8202;
>
> ---
>
> &#8202;
>
> Besides the issues described above, we note the following slight improvement to our work, which we will incorporate into the camera‑ready version of the paper.
>
> &#8202;
>
> -   ___Matching signal classes in Corollary 2 (Remark 2.2, first bullet point).___ Revisiting the proof of the necessity statement of Wang et al. [16, Theorem 2], we verified that the same Fano-based argument applies verbatim to show the necessity statement for binary signals, i.e. $\beta^\star \in \\{0,1\\}^p$. In fact, at a high level, the authors show the necessary condition for $\beta^\star \in C_{p,s}$ by first showing it for $\beta^\star \in \\{0,1\\}^p$ then adding that if no decoder can recover $\beta^\star \in \\{0,1\\}^p$ then no decoder can recover $\beta^\star \in C_{p,s}$ (since $\\{0,1\\}^p \subseteq C_{p,s}$), which weakens their result. This closes the gap between signal spaces, unifying both necessary and sufficient conditions for binary signals. In the camera‑ready version, we will restate the necessary condition of Corollary 2 for binary signals and remove the first bullet point of Remark 2.2.
>
> &#8202;

---

> ### Comment · Reviewer_KJwa · 2025-08-04
>
> Thanks to the authors for their careful replies to my questions. My opinion of the paper remains very positive, and I have no further requests or questions.

---

### Official Review · Reviewer_LvPh · 2025-07-03

**Clarity:** 3
**Significance:** 3
**Originality:** 3
**Rating:** 4
**Confidence:** 3

**Summary:**

This paper studies sparse signal recovery under the sparse measurement matrix assumption. The paper shows an information-theoretic sample complexity threshold for successful sparse recovery, i.e., the phase transition happens at $n^{SP}_{INF}= \Theta(s\log{(p/s)}/\log{(ds/p)})$ where d is the non-zero entries of the measurement matrix, s is the non-zero entries of the signal and p is the dimension of the signal. Furthermore, the paper shows a simplified sample complexity for a specific regime where $s$ and $d$ is asymptotically proportional to $p$.

**Questions:**

Besides the issue I mentioned in above, I would like a summary table that compares the sample complexity for different cases among the previous work and this paper.

**Ethical Concerns:**

["NO or VERY MINOR ethics concerns only"]

**Final Justification:**

About infinity signal to noise ratio, I understand it is non-trivial even for case of infinite signal to noise ratio, I have preference for limited signal to noise ratio. Other comments look good to me. Therefore, I remain my score.

**Limitations:**

Yes

**Quality:**

3

**Strengths And Weaknesses:**

Strength: The problem set up (i.e., the sparse measurement assumption) looks interesting. The result in Theorem 1 looks solid and can be related to previous work.
Weakness: The major concern I have is around the assumption of $\sigma^2=\Theta (1)$ and therefore are dropped in later results (e.g. Corollary 2 and Theorem 3). The noise level $\sigma^2$ plays a very important role in SNR and SNR is the major key component (if not the most important)  that determines how difficult the recovery problem is and thus how much sample would be roughly needed.
 If we assume $\sigma^2=\Theta (1)$, then $SNR \rightarrow \infty$ when $ds/p \rightarrow \infty$ which is an easy task. Previous works that study the linear problem under asymptotic regime mostly need to control $SNR=\Theta(1)$ or at least the results depend on $||\beta^*||$ or $\sigma^2$ so that we can derive the corresponding sample complexity when $SNR=\Theta(1)$. For example, in [13] (referenced in this paper), the abstract points out that "high signal to noise ratio $k/\sigma^2 = \Omega (1)$ regime". In the light of this, I would recommend relaxing the assumption of $\sigma^2=\Theta(1)$ and revising the introduction and the results other than Theorem 1 accordingly.

---

> ### Author Rebuttal · Authors · 2025-07-30
>
> &#8202;
>
> We thank the reviewer for their thorough evaluation. Below, we address their questions and concerns to the best of our ability. At the end, we note a slight improvement to our work that we will incorporate into the camera‑ready version of the paper.
>
> &#8202;
>
> ---
>
> &#8202;
>
> >- **Weaknesses:** The major concern I have is around the assumption of $\sigma^2 = \Theta\left(1\right)$ and therefore are dropped in later results (e.g. Corollary 2 and Theorem 3). The noise level $\sigma^2$ plays a very important role in SNR and SNR is the major key component (if not the most important) that determines how difficult the recovery problem is and thus how much sample would be roughly needed. If we assume $\sigma^2 = \Theta\left(1\right)$ , then $\mathrm{SNR} \rightarrow +\infty$ when $ds/p \rightarrow +\infty$ which is an easy task. Previous works that study the linear problem under asymptotic regime mostly need to control $\mathrm{SNR} = \Theta\left(1\right)$ or at least the results depend on $\left\| \beta^\star \right\|$ or $\sigma^2$ so that we can derive the corresponding sample complexity when $\mathrm{SNR} = \Theta\left( 1 \right)$. For example, in [13] (referenced in this paper), the abstract points out that "high signal to noise ratio regime". In the light of this, I would recommend relaxing the assumption of $\sigma^2 = \Theta\left(1\right)$ and revising the introduction and the results other than Theorem 1 accordingly.
>
>
> &#8202;
>
> - **Rebuttal:** We appreciate the reviewer’s thoughtful concern regarding the assumption that $\sigma^2 = \Theta(1)$, which implies that $\mathrm{SNR} \rightarrow +\infty$  when $ds/p \rightarrow +\infty$. We think of the sparse recovery problem as exhibiting two components of difficulty: a combinatorial component, consisting of identifying the correct support of $\beta^\star$ given a random projection $X \beta^\star$; and a statistical component, consisting of reliably isolating the signal from the noise. Intuitively, the assumptions of $\sigma^2 = \Theta(1)$ and $ds/p \rightarrow +\infty$ allow us to isolate the combinatorial component by making the SNR go to infinity in high dimensions. We note that *even* in absence of noise, namely when SNR is $\infty$, identifying the correct support of the signal remains highly non-trivial, and is known, in the dense design setting, to exhibit phase transitions and fundamental hardness: see Gamarnik and Zadik [7] and Wang et al. [16] in our manuscript, as well as classical papers on compressive sensing such as "Sparse nonnegative solution of underdetermined linear equations by linear programming" by Donoho and Tanner and "Decoding by linear programming" by Candes and Tao.
>    We agree that assuming a finite or vanishing SNR would make the setting more realistic in some applications. In that case, the recovery problem becomes more complex: one must not only identify the correct support, but also reliably isolate the signal from the noise. While this combined regime is indeed of interest, it is at this point beyond the scope of this paper. We believe our work provides a step toward extending these results to the finite-SNR regime, which is a valuable direction for future work. On the other hand, we acknowledge that it is important for the paper to explicitly mention that we are considering the high-SNR regime, and we appreciate the reviewer’s observation in this regard. We will clarify this point in the camera-ready version of the paper.
>
> &#8202;
>
> ---
>
> &#8202;
>
> >- **Questions:** Besides the issue I mentioned in above, I would like a summary table that compares the sample complexity for different cases among the previous work and this paper.
>
>
> &#8202;
>
> - **Rebuttal:** We thank the reviewer for this helpful suggestion. We fully agree that a summary table comparing the sample complexity across different settings would improve the paper. In the camera-ready version, we will add the following table summarizing the key information-theoretic and algorithmic sample complexity results from prior works alongside the new results presented in this paper.
>    &#8202;
> | Regime | Information-theoretic, Necessary | Information-theoretic, Sufficient | Algorithmic, Necessary | Algorithmic, Sufficient |
> |---|---|---|---|---|
> | Dense $(d=p)$ | $\frac{2 s \log(p/s)}{\log{s}}$ ([13], [16]) | $\frac{2 s \log(p/s)}{\log{s}}$ ([7], [13]) | $2 s \log(p-s)$ ([7]) | $2 s \log(p-s)$ ([15]) |
> | Sparse | $\frac{2 s \log(p/s)}{\log(ds/p)}$ (Corollay 2, originally by [16]) | $\frac{2 s \log(p/s)}{\log(ds/p)}$ (Theorem 1) | No result known so far |$2 s \log(p-s)$ ([12])|
>
> &#8202;
>
> ---
>
> &#8202;
>
> Besides the issues described above, we note the following slight improvement to our work, which we will incorporate into the camera‑ready version of the paper.
>
> &#8202;
>
> -   ___Matching signal classes in Corollary 2 (Remark 2.2, first bullet point).___ Revisiting the proof of the necessity statement of Wang et al. [16, Theorem 2], we verified that the same Fano-based argument applies verbatim to show the necessity statement for binary signals, i.e. $\beta^\star \in \\{0,1\\}^p$. In fact, at a high level, the authors show the necessary condition for $\beta^\star \in C_{p,s}$ by first showing it for $\beta^\star \in \\{0,1\\}^p$ then adding that if no decoder can recover $\beta^\star \in \\{0,1\\}^p$ then no decoder can recover $\beta^\star \in C_{p,s}$ (since $\\{0,1\\}^p \subseteq C_{p,s}$), which weakens their result. This closes the gap between signal spaces, unifying both necessary and sufficient conditions for binary signals. In the camera‑ready version, we will restate the necessary condition of Corollary 2 for binary signals and remove the first bullet point of Remark 2.2.
>
> &#8202;

---

### Note · Authors · 2025-08-13

&#8202;

We thank the reviewers for their insightful observations. As a conclusion to the discussions we had, we provide below a summary of the changes that we will make to the camera-ready version of the paper, taking advantage of the extra one-page allowance.  We will:

&#8202;

- **Matching signal classes in Corollary 2 (Remark 2.2, first bullet point):**  Prove that the necessity statement shown by Wang et al. [16] extends to the binary setting, and update Corollary 2 to state the necessity result for binary signals. Remove the first bullet of Remark 2.2.

&#8202;

- **Inconsistency of the notions of "reliable recovery" in necessity and sufficiency statements (Remark 2.2, second bullet point):**  Revise the second bullet point of Remark 2.2 to correctly describe Reeves et al. [13] and discuss the differences between our work and theirs.


&#8202;

- **Description of the "High $\text{SNR}$" regime:** Add the description of "High $\text{SNR}$" or "$\text{SNR} \rightarrow +\infty$" to mentions of the "regime where $ds/p \to +\infty$" in the abstract as well as sections 1, 2, 4.

&#8202;

- **Summary table for sample complexity:** Add a table summarizing prior results as well as ours, on sample complexity in each of the dense and sparse settings.

&#8202;

- **Description of the sparsification setting:** Frame the sparsification setting (Section 3) as using a different matrix for measurement versus recovery.

&#8202;

- **Highlighting the importance of support recovery:** Note its equivalence to the squared $\ell_2$-loss in the binary case and MLE recovery once the support is known.

&#8202;

- **Clarifying notation:** We will clarify in the abstract that $n_{\mathrm{INF}}^{\mathrm{SP}}$ denotes the number of measurements and add a “Notation” paragraph at the end of the Introduction to define symbols such as the entropy function $H(\cdot)$ and the $\ell_0$-norm $||\cdot||_0$.

&#8202;

- **Clarifying sparsity definition:** Explicitly state that, in the extended setting described in lines 67-68, "sparsity" means having $s$ non-zero components each of magnitude at least $\lambda$, replacing the binary constraint.

&#8202;

- **Acknowledging related missing-data literature:** Add a discussion citing Loh and Wainwright [2012] and related work, and explaining how it relates to our setting.

&#8202;

---

### Decision · Program_Chairs · 2025-09-17

**Decision:**

Accept (poster)

**Comment:**

### Summary
This paper studies sufficient conditions for successful support recovery of a $p$-dimensional $s$-sparse signal using its noisy linear measurement with a sparse matrix (with $d$ non-zero elements per row). Together with the necessary condition reported in [16] (reproduced as Corollary 2 (i) in this paper), the result of this paper states that in the two regimes $ds/p\to\infty$ and $ds/p\to\tau\in(0,\infty)$ the problem exhibits a phase transition at an information-theoretic criticality in the number of samples. As a second contribution, this paper stidies support recovery of a sparse signal using noisy linear measurement with a dense matrix, while a *sparsified* version of the measurement matrix is to be used in recovery. A sufficient condition for successful support recovery for this setting (Theorem 3) is obtained, under a certain integrability conjecture.
### Strengths
- The subject is clearly of interest to the NeurIPS community.
- The problem setting of considering a sparse measurement matrix is of practical significance.
- This paper is well written and well organized.
### Weaknesses
- There remain small technical gaps (difference in the signal models, as well as exact versus approximate recovery, as well as integrability conjecture), which have been explicitly mentioned in the main text (Remarks 2.2, 3.1).
### Reasons for decision
The rating/confidence scores from the four reviewers were 4/3, 5/4, 5/4, and 6/4, all being on the positive side, with the average 5 well above the threshold of acceptance. A majority of the review comments were clarifying ones, and overall, the author response seems to have successfully addressed those comments on unclear points raised by the reviewers. Because of these, I am glad to recommend acceptance.
### Additional points
- Line 162: the set (of) supports
- References:
  - Capitalization is not coherent throughout the reference list.
  - [9]: The bibliographic information seems imcomplete.
  - [15]: using (*\\ell*\_\{1\} → $\ell_1$)-constrained